# Inhibition of UBA6 by inosine augments tumour immunogenicity and responses

Lei Zhang[1,2,18], Li Jiang[3,18], Liang Yu[4,18], Qin Li[5,18], Xiangjun Tian[6], Jingquan He[7], Ling Zeng[1,2], Yuqin Yang[8], Chaoran Wang[5], Yuhan Wei[5], Xiaoyue Jiang[5], Jing Li[9], Xiaolu Ge[1,2], Qisheng Gu [4], Jikun Li[4], Di Wu[10,11], Anthony J. Sadler[12,13], Di Yu [14], Dakang Xu[15], Yue Gao [16] ✉, Xiangliang Yuan [15] ✉ & Baokun He[1,2,17] ✉

Anti-cancer immunity and response to immune therapy is influenced by the metabolic states of the tumours. Immune checkpoint blockade therapy (ICB) is known to involve metabolic adaptation, however, the mechanism is not fully known. Here we show, by metabolic profiling of plasma samples from melanoma-bearing mice undergoing anti-PD1 and anti-CTLA4 combination therapy, that higher levels of purine metabolites, including inosine, mark ICB sensitivity. Metabolic profiles of ICB-treated human cancers confirm the association between inosine levels and ICB sensitivity. In mouse models, inosine supplementation sensitizes tumours to ICB, even if they are intrinsically ICB resistant, by enhancing T cell-mediated cytotoxicity and hence generating an immunologically hotter microenvironment. We find that inosine directly inhibits UBA6 in tumour cells, and lower level of UBA6 makes the tumour more immunogenic and this is reflected in favourable outcome following ICB therapy in human melanomas. Transplanted mouse melanoma and breast cancer cells with genetic ablation of *Uba6* show higher sensitivity to ICB than wild type tumours. Thus, we provide evidence of an inosine-regulated UBA6-dependent pathway governing tumour-intrinsic immunogenicity and hence sensitivity to immune checkpoint inhibition, which might provide targets to overcome ICB resistance.

Cancer immunotherapy has become highly successful against cancers by triggering the cytotoxic potential of the human immune system. Among cancer immunotherapy, immune checkpoint blockade (ICB), which targets cytotoxic T lymphocyte antigen 4 (CTLA4) or the programmed cell death 1 (PD1)-ligand 1 (PD L1) axis has been approved for treating many different cancers[1]. Despite some impressive clinical outcomes from ICB, most patients still do not obtain a meaningful response to ICB, and the key drivers of this heterogeneity are not fully understood. Tumour-intrinsic factors (such as low mutational burden and local immunosuppression) and host-related factors (such as age, hormones, and genetic polymorphisms) may contribute to this heterogeneity of response to ICB[2,3].

Recently, various independent studies in both mice and humans have highlighted that gut microbiota affects the outcomes of ICB[4–10] and the particular commensal species have been identified to be associated with beneficial clinical response to ICB[5–7]. However, although some general trends were independently observed[5–7], the consistent specific causal microbial taxa didn't converge in ICB responders, thus mechanisms underlying this relationship remain unclear. Notably, the molecular mechanism whereby gut microbiota influences immune responses is mainly assigned to gut microbial metabolites, such as short-chain fatty acid and bile acids, which play a critical role in immune homoeostasis and influence the susceptibility of the host to immune response[11–13]. It also has been well established that the tumour microenvironment (TME) has been significantly

changed by altered tumour energy metabolism, leading to the enhanced metabolites which have been identified to serve as signalling molecules in modulating immunity[14–16]. Small metabolites, derived from microbiota or metabolic TME, not only are essential intermediates in intracellular biochemical processes but can also influence neighbouring cell functions[11,16–18]. Interestingly, there is growing evidence that metabolic alterations of cancer cells or microbiota modulate tumour immunity by regulating immune cells in the tumour immune microenvironment[14,19–22]. However, the detailed understanding of the effects of metabolic alterations on immunotherapy responses has remained exclusive. We hypothesised that metabolites can serve as environmental cues, and mediate antitumour immune responses to influence the immunotherapy efficacy of cancer.

Here we identify, by employing untargeted metabolomics analyses of plasma from large cohorts of cancer patients and mouse tumour models undergoing ICB treatment, metabolites which are associated with responses to ICB. Notably, we find the metabolite inosine to enhance ICB responses. Mechanistically, inosine mainly targets tumour cells rather than immune cells, hence augmenting tumour immunogenicity to overcome tumour-intrinsic resistance to ICB. We identify the direct target of inosine within the tumour cells in mice and in cancer patients, and validate its function in mouse models by genetic approaches.

## Results

### A metabolic screen identifies inosine is associated with immunotherapy responses in mice and humans

To better understand the association between metabolic alterations and ICB responses, we performed untargeted metabolomics of plasma samples from B16-F0 tumour-bearing mice with vehicle or ICB (anti-PD1 plus anti-CTLA4) treatment (Supplementary Fig. 1a). The metabolic profiling revealed that the relative abundance of 5.3% (13/244) metabolites were significantly altered in B16-F0 tumour-bearing mice with ICB treatment (Supplementary Fig. 1b, c and Data 1). Notably, 5 of these 13 changed metabolites were involved in purine metabolism, including inosine, guanosine, hypoxanthine, and xanthine (Supplementary Fig. 1a–c and Data 1). Given the role of gut microbiota in purine metabolism and immunotherapy[7,23,24], we performed the depletion of gut microbiota with an antibiotic cocktail (Abx) which not only significantly compromised the efficacy of ICB in the B16-F0 mouse model but also decreased the levels of these purine metabolites (Supplementary Fig. 1d–f), indicating that disorders of purine metabolism induced by ICB may be partly due to gut microbiota dysbiosis. To further reveal gut microbiota dysbiosis induced by ICB treatment, we used 16 S rRNA gene sequencing to determine the microbiota composition in stool samples from the B16-F0 tumour-bearing SPF mice with Ctrl or ICB treatment. Consistent with prior reports in mice and humans[7,9], ICB therapy led to altered microbial composition in the gut in the B16-F0 mouse model following ICB therapy (Supplementary Fig. 2a–d). Partial least squares discriminant analysis (PLS-DA) showed that the overall microbial community in B16-F0 tumour-bearing SPF mice with Ctrl treatment was completely separated from that in B16-F0 tumour-bearing SPF mice with ICB treatment (Supplementary Fig. 2a). Although a minor change in microbiota composition at the phylum and genus level in B16-F0 tumour-bearing SPF mice with ICB treatment, the relative abundance of some genera, such as *Parabacteroides*, *Akkermansia*, *Bifidobacterium*, were moderately changed by ICB treatment (Supplementary Fig. 2b–d). Notably, Mager et al. recently reveal that some species of *Akkermansia* and *Bifidobacterium* enhance the response to CTLA4 antibody by producing large amounts of inosine[25]. Collectedly, these findings indicate that the changes in plasma levels of purine metabolites, especially inosine, might at the least be partially due to gut microbiota dysbiosis induced by ICB treatment.

Moreover, we analysed the metabolic profiling of renal cell carcinoma (RCC) patients, among which 394 received nivolumab, a PD1 checkpoint blockade, and 349 received everolimus, an mTOR inhibitor (Phase III trial: CheckMate 025, NCT01668784)[15,26]. The results showed that the plasma levels of 37/202 metabolites were associated with overall survival (OS) of cancer patients treated with nivolumab ($P < 0.01$) (Fig. 1a and Supplementary Data 2). Notably, Venn diagram analysis demonstrated that only inosine was significantly associated with ICB response in both mice and humans (Supplementary Fig. 1e). Specifically, the higher level of inosine was associated with the longer OS of cancer patients only in the setting of nivolumab treatment (high: mOS = 33 months; low: mOS = 22 months), but not in cancer patients treated with everolimus (an mTOR inhibitor) (Fig. 1b), indicating that high level of inosine had a durable benefit for ICB-treated patients.

Indeed, the B16-F0 model had high sensitivity to ICB treatment (Supplementary Fig. 1a). Interestingly, plasma levels of inosine in B16-F0 tumour-bearing mice were higher than that in tumour-free mice (Supplementary Fig. 2f). However, plasma levels of inosine were reduced in both tumour-bearing mice and cancer patients after ICB treatment (Supplementary Figs. 1c, 2g). Collectively, these findings indicate that metabolite inosine is associated with immunotherapy responses and patients with a higher level of inosine may benefit from ICB treatment.

### Inosine augments ICB immunotherapy responses in vivo

The strong association between inosine level and ICB responses suggests a potential role of inosine in enhancing immune response. This prompted us to investigate whether systemic administration of inosine could augment immunotherapy response in vivo. Although inosine has been used as a dietary supplement or immunomodulatory drug for several decades[27], its application in cancer immune therapies remains exclusive. Indeed, inosine alone significantly reduced tumour growth in the B16-F0 model (Fig. 1c). Strikingly, mice in the combined inosine with ICB treatment regimen (Combo) had the best response in the B16-F0 model (Fig. 1c). Next, we moved forward to assess the efficacy of inosine in combination with ICB in the B16-GMCSF model, which is resistant to ICB[28]. Surprisingly, the combo treatment overcame the resistance to ICB and resulted in the elimination of >80% of B16-GMCSF tumours (Fig. 1d and Supplementary Fig. 3a–c). Most importantly, the combo treatment increased the OS of B16-GMCSF tumour-bearing mice in comparison with either inosine or ICB alone (Fig. 1d).

To test whether the synergistic effect in the B16-GMCSF model also extends to other ICB-resistant models, we evaluated the efficacy of inosine and ICB combination therapy in the 4T1 tumour model (murine triple-negative mammary carcinoma in Balb/c background), which was aggressive and highly resistant to ICB treatment. Consistent with the B16 melanoma models, inosine and ICB combination therapy promoted long-term survival of 4T1 tumour-bearing mice and led to complete remissions in 50% of 4T1 tumour-bearing mice (Fig. 1e and Supplementary Fig. 3d–f). Notably, no body weight loss was observed after treatment with Ino or Ino+ICB in the 4T1 model (Supplementary Fig. 3g). We also measured the serum levels of alanine aminotransferase (ALT) and aspartate aminotransferase (AST) to reveal no signs of hepatotoxicity in the 4T1 tumour-bearing mice with Ino or Ino+ICB treatment (Supplementary Fig. 3h). Treatment with Ino or Ino+ICB did not induce a significant inflammation of lung and liver in 4T1 tumour-bearing mice (Supplementary Fig. 3i), confirming that inosine does not induce systemic inflammation. Moreover, we identified the synergistic efficacy of isoprinosine, an inosine derivative, in combination with ICB in the 4T1 model (Supplementary Fig. 3j–l). Collectively, given inosine is a safe, naturally occurring purine with non-toxic to humans, coupled with our preclinical evidence showing its synergic effect with ICB, it is worthwhile to repurpose the therapeutic potential of inosine for enhancing cancer patient response to immunotherapies.

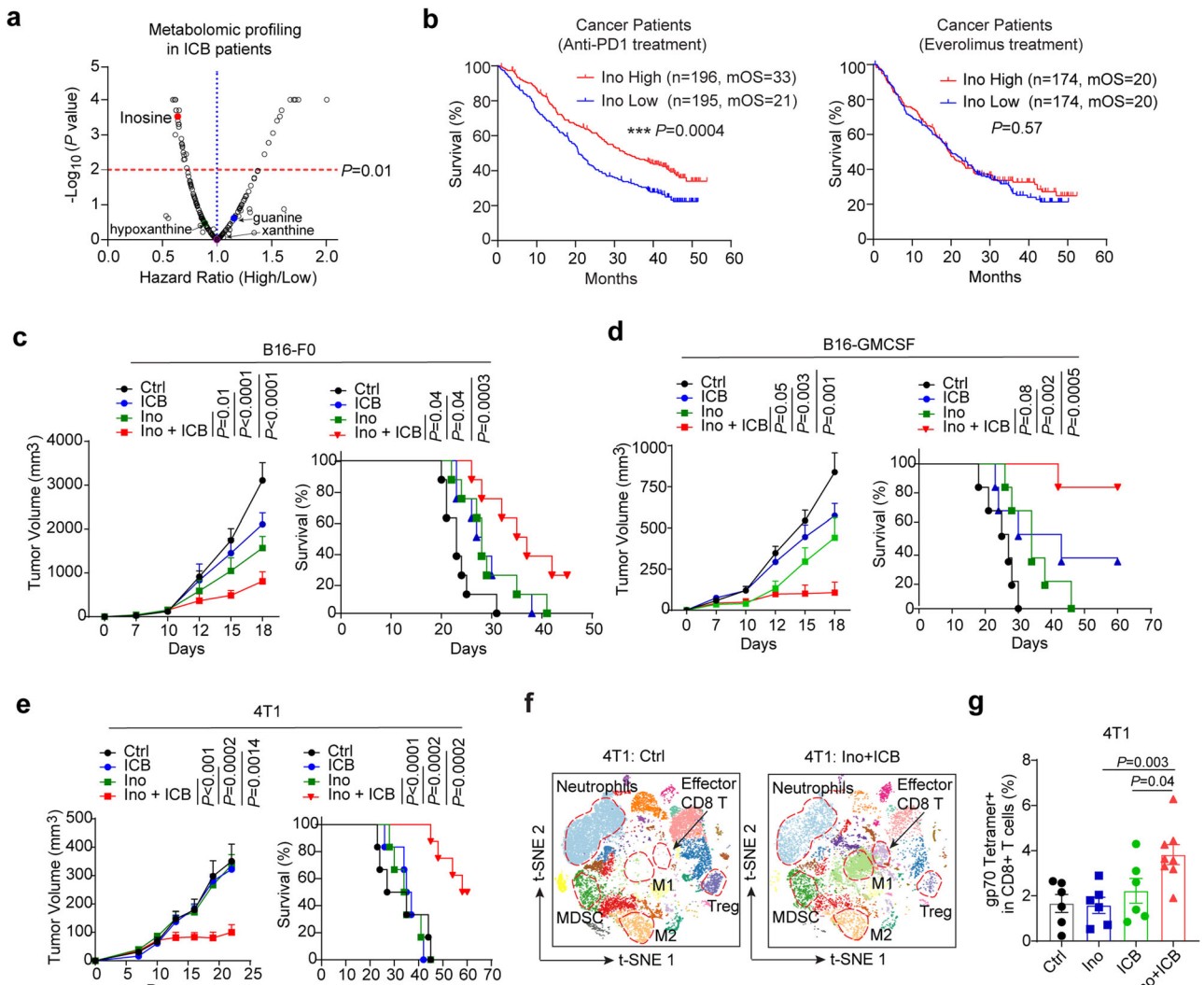

**Fig. 1 | Inosine overcomes resistance to immunotherapy by inflaming tumour immune microenvironment. a** Volcano plot showing the hazard ratios (HR) of high/low levels of serum metabolites (*n* = 202, represented as points, high and low are stratified by the median of each metabolite) in CheckMate 025 renal cell carcinoma (RCC) patients with nivolumab (anti-PD1 Ab) treatment (*n* = 392). Overall survival is used as a Cox proportional hazards model. A cutoff at *P* = 0.01 is shown as a horizontal line and HR (high/low) = 1 is shown as a vertical line (HR: high/low <1 indicates that a high level of a metabolite is a benefit for ICB patients). **b** Kaplan–Meier plot of overall survival in RCC patients with nivolumab (anti-PD1 Ab) (High, *n* = 196, mean OS = 33; Low, *n* = 195, mean OS = 21) or everolimus (mTOR inhibitor) (High, *n* = 174, mean OS = 20; Low, *n* = 174, mean OS = 20) grouped by the

inosine level at the median of baseline level. **c**–**e** Tumour volume and survival analysis of B16-F0 (**c**, *n* = 5), B16-GMCSF (**d**, *n* = 5), or 4T1 (**e**, *n* = 6) tumour-bearing mice treated with IgG2a (Ctrl), 400 mg/kg of Inosine (Ino), anti-CTLA4 + anti-PD1 (ICB) or ICB + Inosine (Ino + ICB) treatment. **f** tSNE plot of single-cell RNA sequencing of CD45+ immune cells from 4T1 tumours treated with Ctrl (*n* = 16199 cells) or Ino+ICB treatment (*n* = 9842 cells). **g** The frequency of gp70-specific CD8+ T cells in 4T1 tumour with Ctrl (*n* = 6), Ino (*n* = 6), ICB (*n* = 6), or Ino+ICB (*n* = 8) treatment for 15 days. Data are presented as Mean ± s.e.m. Statistical significance was determined by one-way ANOVA and Tukey test for multiple comparisons (tumour sized of **c**–**e**, **g**) or log-rank (Mantel–Cox) test (survival analysis of **b**–**e**). Source data are provided as a Source Data file.

## Inosine inflames the tumour immune microenvironment (TIME)

To provide a more comprehensive and unbiased assessment of the effect of inosine on TIME, single-cell RNA sequencing (scRNA-seq) of CD45+ immune cells in the 4T1 model was performed. We obtained single-cell transcriptomes for 16,199 CD45+ cells in the control group, 9842 in the Combo group. To define the intratumoural cell populations, we performed canonical correlation analysis to computationally combine data from two treatment groups, then conducted graph-based clustering and dimensionality reduction with UMAP to respectively identify and visualise transcriptionally homogeneous clusters of immune cells (Fig. 1f and Supplementary Fig. 3m, n). Using the SingleR package, we further annotated the clusters by directly comparing their transcriptional state with that of known populations and the assessment of cell-type-specific markers[29]. We compared the immune microenvironment of combo and control 4T1 tumours and

found a significant increase in Ki67+ CD8+ T cells in combo-treated tumours that were infiltrated throughout the tumour with an increase in CD8+/Treg ratio following combinational treatments, as determined by manual gating analyses and reflect the induction of an effective immune response by inosine in combination with ICB (Supplementary Fig. 3n). Moreover, analysis of myeloid cells in the combination regimen of inosine with ICB showed that the addition of inosine reduces the immunosuppressive microenvironment by increasing the M1/M2 ratio, resulting in improved T-cell effector function in 4T1 tumours (Supplementary Fig. 3m, n). Notably, the addition of inosine with ICB treatment caused a striking shift of immunosuppressive to inflammatory TIME characterised by the decreased accumulation of M2 macrophages and Tregs, and the increased abundance of M1 macrophages and effector CD8+ T cells (Fig. 1f and Supplementary Fig. 3m, n). Specifically, a substantial

increase in tumour-reactive gp70 tetramer-specific CD8[+] T cells in 4T1-bearing mice with combo treatment was also identified by flow cytometric analysis ($P < 0.01$; Fig. 1g and Supplementary Fig. 3o), indicating a strong specific antitumour immunity after addition of inosine. Thus, these findings suggest that inosine in combination with ICB inflames TIME to provoke a strong antitumour immune response.

## Inosine sensitises tumour cells to T-cell-mediated killing by enhancing tumour-intrinsic immunogenicity

To reveal the mechanism by which inosine influences antitumour immunity, we set up the different strategized in vitro co-culture platforms of T-cell-mediated tumour cell killing assay to evaluate the effect of inosine on tumour cells and T cells simultaneously. Despite previous reports have indicated the multiple immunomodulatory roles of inosine on immune cells under different conditions[27,30], we did not find stronger T-cell-mediated tumour killing in B16-GMCSF and 4T1 tumour cells when we pretreated T cells with inosine compared to untreated control (Fig. 2a–c). However, when we pretreated tumour cells with inosine and then co-cultured tumour cells with activated T cells, we found that both 4T1 and B16-GMCSF tumour cells were dramatically sensitive to T-cell-mediated cytotoxicity, as indicated by the lower cell viability in the inosine-pretreated group compared to unpretreated control (Fig. 2d, e), suggesting the direct effect of inosine on tumour cells.

Notably, inosine didn't directly influence the proliferation and apoptosis of 4T1 cells (Fig. 2f, g) or B16-GMCSF cells (Fig. 2f, h). Importantly, we further identified that inosine markedly potentiated MHC-I upregulation (Fig. 2i). In addition, inosine treatment increased the expression of related genes involved in antigen processing/presentation and IFN-γ responses in 4T1 or B16-GMCSF tumour cells (Fig. 2j, k), establishing the functional importance of inosine on tumour cell immunogenicity. Thus, our data indicate that inosine renders tumour cells more sensitive to T-cell-mediated tumour killing by directly modulating tumour cell immunogenicity.

## Inosine directly inhibits UBA6 activity in tumour cells

To directly explore by which inosine elicits tumour immunogenicity, chemical proteomics screening following a LiP-small molecule mapping (LiP-SMap) workflow[31] in 4T1 cell lysate was performed to identify the functional proteins potentially binding with inosine in tumour cells (Fig. 3a). Significant changes in the abundance of half-tryptic peptides (fold change (FC) >2 or <0.5, $p < 0.001$, >2 peptides per protein) were read out for structural changes induced by the binding of inosine. Out of 2470 proteins, only 23 proteins fulfilled these stringent criteria (Fig. 3b, c and Supplementary Fig. 4a, Data 3).

We further identified which candidates bind with inosine involved in immune cell-mediated tumour killing. The gene knockout phenotype from genetic screens profiling regulators of lymphocyte mediated tumour killing resistance based on several CRISPR genetic screen data sets was analysed[32]. Out of 23 candidates, only *Uba6* deletion in tumour cells enhanced the T-cell[33] (Fig. 3d and Supplementary Fig. 4b) or NK cell-mediated tumour killing[34] (Supplementary Fig. 4c). Moreover, we developed a pooled genetic screening approach to identify 23 genes that may increase or decrease the fitness of 4T1 tumour cells growing in vivo. After 12 days, we collected the tumours and compared the library representation in tumours from WT mice to tumours growing in NSG mice. Our results revealed that UBA6 deletion, among the 23 identified genes, had the highest negative score which indicates the increased sensitivity of tumour cells to immune attack in vivo (Supplementary Fig. 4d), consistent with the results from several CRISPR genetic screen data sets (Fig. 3d and Supplementary Fig. 4b, c). Altogether, these findings indicate that UBA6 may play a critical role in the effect of inosine on immunotherapy responses.

UBA6, ubiquitin-like modifier activating enzyme 6, is one of the ubiquitin-activating enzymes which activates and transfers the ubiquitin to the subsequent proteins to serve as the starting enzyme for the extensive downstream ubiquitination cascades[35]. Besides, UBA6 also activates the ubiquitin-like proteins FAT10 and transfers FAT10 to its substrate proteins, leading to its proteasomal degradation independently from ubiquitin[36]. Owing to the central role in UBA6-dependent post-translational modification, UBA6 participates in multiple pathogeneses of diseases. However, how inosine regulates UBA6 activity to modulate immunotherapy is unclear.

To further explore the regulation of inosine on UBA6 activity, we measured the impact of inosine on interactions between UBA6 and USE1 proteins. Our results indicated that inosine reduced the interactions between UBA6 and USE1 (UBA6-specific ubiquitin E2) in HEK293 cells (Fig. 3e). We also used the purified UBA6 and USE1 proteins in a cell-free assay to confirm that the preincubation of UBA6 with inosine directly reduced the interactions of UBA6 with USE1 (Supplementary Fig. 4e). Considering the bispecific effect of UBA6 on ubiquitin and FAT10 using a similar mechanism with greater affinity for FAT10[37], we used the purified UBA6 and USE1 proteins to set up thioester activity assay in vitro. Our results showed that inosine decreased the UBA6-mediated FAT10 thioester of USE1 protein in vitro (Fig. 3f). It was previously reported that FAT10-dependent degradation machinery was linked to antigen processing pathway and inflammatory signalling pathway[38,39]. Inosine had a modest effect on the ubiquitination of total proteins (Supplementary Fig. 4f), which is probably due to the responsibility of UBA6 for charging <1% of ubiquitination[40]. We found that inosine exhibited a moderate effect on UBA6-mediated transfer ubiquitin of USE1 protein in vitro (Supplementary Fig. 4g). Moreover, the deletion of UBA6's ubiquitin-fold domain (UFD), which is responsible for the interaction between UBA6 and USE1, led to the loss of function of UBA6 on USE1 ubiquitination and abolished the effect of inosine on the interaction between UBA6 and USE1 in HEK293 cells (Supplementary Fig. 4h, i). Functionally, loss of UBA6 in tumour cells sensitised tumour cells to the cytotoxicity of T cells and abolished the effect of inosine on T-cell-mediated tumour killing (Fig. 3g). Collectively, our results indicate that inosine sensitises tumour cells to T-cell-mediated killing by directly inhibiting UBA6 activity.

## Inosine and genetic inhibition of UBA6 increase tumour immunogenicity

To decipher the molecular mechanism of the UBA6 effect on tumour cells, we analysed the transcriptome of *Uba6*-null 4T1 cells and WT 4T1 cells by RNA-seq (Supplementary Data 4). Remarkably, enrichment analysis revealed that *Uba6*-null tumour cells had a marked increase in gene expression profiles evoked by inflammatory cytokines, such as TNF-α, IFNα, and IFNγ (Fig. 4a–c and Supplementary Fig. 5a, b, Data 5). The qPCR analysis confirmed the upregulation of TNF-α and IFN response genes, and antigen presentation-related genes in *Uba6*-null 4T1 cells (Supplementary Fig. 5c–e).

In addition, proteomic profiles also confirmed the higher engagement of the IFN signalling pathway and inflammatory response signalling in *Uba6*-null tumour cells, showing consistency between our proteomic and transcriptomic data sets (Fig.4d and Supplementary Fig. 5f). The flow analysis proved the upregulated cell surface MHC-I protein expression in *Uba6*-null tumour cells (Supplementary Fig. 5g), which was consistent with the effect of inosine.

Notably, *Uba6* deletion in 4T1 cells reversed the effect of inosine on the expression of immune response-related genes (Fig. 4e), confirming the inhibitory effect of inosine on UBA6 in tumour cells. Although inosine has been reported to execute some of its functions by adenosine receptors (ARs), which are a member of the large family of seven transmembranes spanning G protein-coupled receptors (GPCR)[41,42], ARs rarely expressed in 4T1 and B16-GMCSF tumour cells and were not observed by Lip-SMapapproach (Supplementary

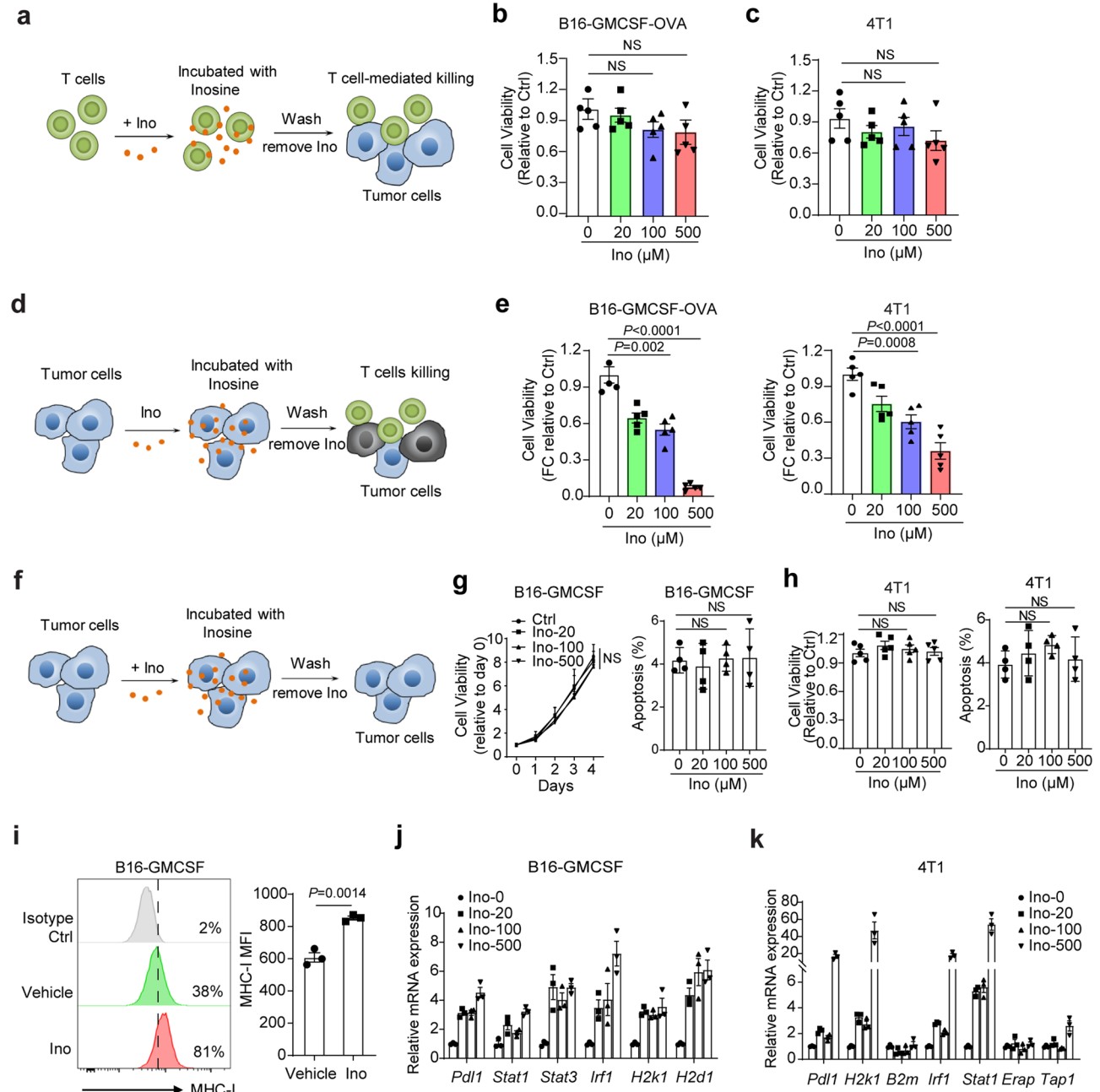

**Fig. 2 | Inosine sensitises tumour cells to T-cell-mediated killing by modulating tumour immunogenicity. a** Experimental strategy to evaluate the ability of inosine to enhance T-cell-mediated tumour killing by modulating T cells. **b** The relative cell viability of B16-GMCSF-OVA cells was shown. OT-1 T cells were pretreated with indicated concentrations of inosine or vehicle for 24 h, then co-cultured with B16-GMCSF-OVA tumour cells at a 2:1 E: T (Effector: T cells, Target: tumour cells) ratio for 48 h ($n = 5$). **c** The relative cell viability of 4T1 cells after incubation with activated CD8⁺ T cells pretreated with the indicated concentration of inosine at a 5:1 E: T ratio for 48 h ($n = 5$). **d** Experimental strategy to evaluate the ability of inosine to enhance T-cell-mediated tumour killing by modulating Tumour cells. **e** The relative cell viability of B16-GMCSF-OVA cells (left) and 4T1 cells (right) with Ctrl or inosine treatment following the method in **a** ($n = 5$). **f** Experimental strategy to evaluate the direct effect of inosine on tumour cells. **g** The relative cell viability (left) and

apoptosis (right) of B16-GMCSF cells following inosine treatment at indicated concentrations for 48 h in vitro ($n = 5$). **h** The relative cell viability (left) and apoptosis (right) of 4T1 cells following inosine treatment at indicated concentrations for 48 h in vitro ($n = 5$). **i** Representative flow analysis (left panel) and quantifying (left panel) the intensity of cell surface MHC-I expression in B16-GMCSF cells treated with vehicle and inosine (100 μM) upon IFN-γ (20 ng/ml) treatment ($n = 3$). **j, k** Selective represented antigen processing/presentation and interferon-responsive gene expression in B16-GMCSF (**j**) and 4T1 (**k**) and tumour cells treated with inosine at indicated concentrations ($n = 3$). Data are presented as Mean ± s.e.m. Statistical significance was determined by Two-sided Student's t test (**i**) and one-way ANOVA and Tukey test for multiple comparisons (**b, c, e, g, h**). NS no significant. Source data are provided as a Source Data file.

Fig. 5h, Data 4), suggesting the effect of inosine on tumour cells, especially 4T1 and B16-GMCSF cells, might not be mediated through ARs. To further reveal the mechanism of how inosine impacts tumour immunogenicity, we used adenosine receptors antagonist CGS15943, which inhibits ARs downstream pathway, and inosine transport ENT1/

ENT2 inhibitor Dilazep dihydrochloride, which blocks extracellular inosine into cells, to pretreat 4T1 cells and then measured the direct downstream gene signature of UBA6. We found that CGS15943 had no impact on inosine-induced function. However, the Dilazep dihydrochloride significantly blocked the expression of several UBA6-

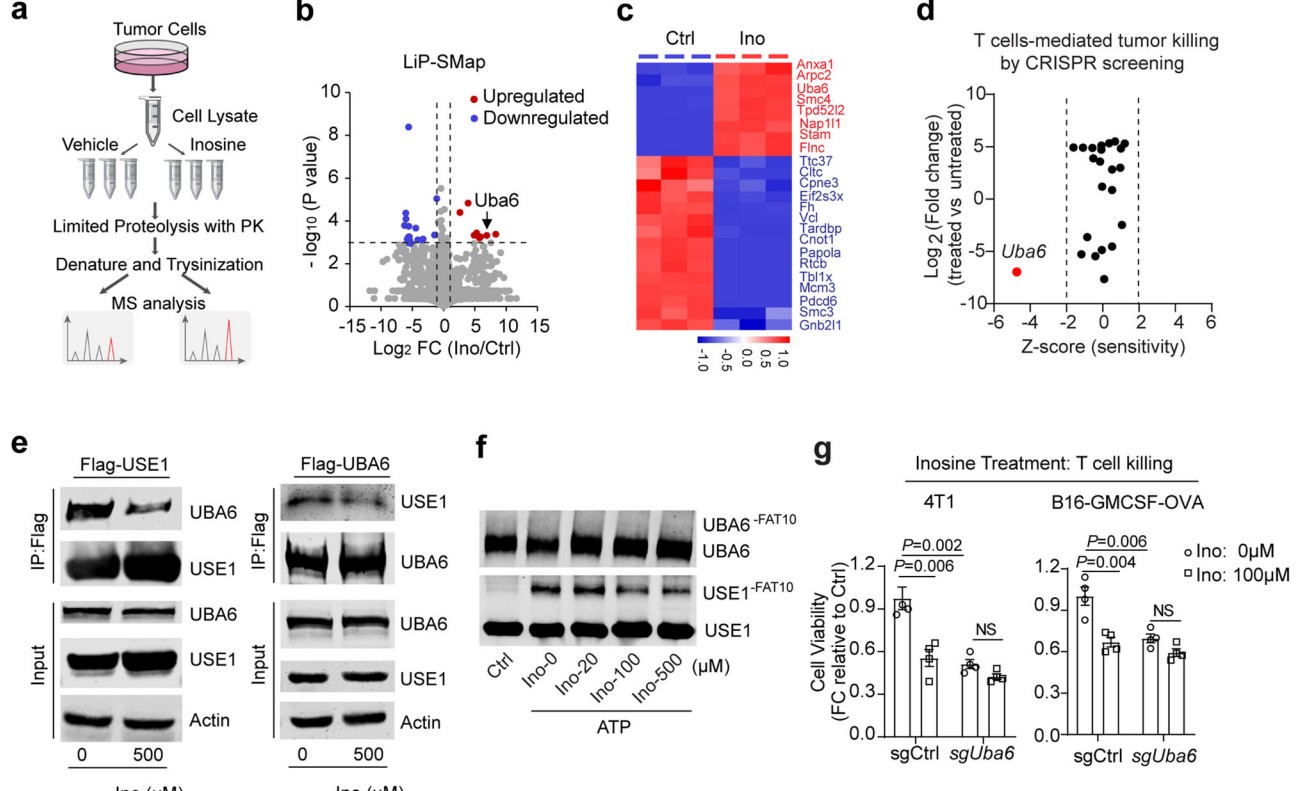

**Fig. 3 | Inosine directly inhibits tumour UBA6 to sensitise T-cell-mediated killing. a** Scheme of chemical proteomics for target identification. **b** Volcano plots of LiP-SMap experiments with inosine treatment. **c** Heat-map of the top 23 proteins changed with inosine treatment identified by LiP-SMap. **d** The effect of deletion of the top 23 genes in **e** on OT-1 T-cell-mediated tumour cell killing. **e** The effect of inosine on the interaction between UBA6 and USE1 in HEK293 cells. Data were representative of two independent experiments ($n = 2$). **f** The effect of inosine on

USE1-S-FAT10 thioester in vitro. Data were representative of two independent experiments ($n = 2$). **g** The relative cell viability of WT (sgCtrl), *Uba6*-null (sg*Uba6*) 4T1, or B16-GMCSF-OVA tumour cells with vehicle (Ctrl) or inosine (100 μM) following the method in **a** ($n = 5$). Data are presented as Mean ± s.e.m. Statistical significance was determined by one-way ANOVA and Tukey test for multiple comparisons (**g**). The P values in b and d were based on log-rank tests. NS no significant. Source data are provided as a Source Data file.

downstream genes induced by inosine treatment (Supplementary Fig. 5i). These results suggest the effect of inosine on tumour immunogenicity is likely dependent on intracellular inosine transported by ENT transporters, but not extracellular receptor ARs. Together, our findings indicate that tumour cell-intrinsic proteins, especially UBA6, play a critical role in the effect of inosine on tumour immunogenicity.

Functionally, *Uba6*-null 4T1 and B16-GMCSF cells showed markedly decreased cell viability when stimulated with TNFα and IFNγ (Fig. 4f and Supplementary Fig. 5j), which mimics the functional feature of inosine on T-cell-mediated tumour killing by targeting tumour cells, as TNF and IFNγ are major cytolytic cytokines released by cytotoxic CD8+ T cells. Altogether, UBA6 deletion in tumour cells primed tumour cell-intrinsic immune response and ablated the effect of inosine on gene expression of immune response signalling.

## UBA6 loss substitutes the effect of inosine on immunotherapy response in vivo

We subsequently assessed the role of UBA6 on the synergistic efficacy of inosine in combination with ICB in vivo. *Uba6* deficiency in B16-GMCSF cells did not markedly affect tumour growth and survival in NSG mice, and *Uba6*-null B16-GMCSF cells did not show any growth disadvantage in vitro (Fig. 5a and Supplementary Fig. 6a, b). By contrast, the *Uba6*-null B16-GMCSF tumour showed a reduced tumour volume and improved survival in WT mice (Fig. 5a and Supplementary Fig. 6a). However, the effect of inosine in combination with ICB on tumour growth was abolished in *Uba6*-null B16-GMCSF tumour-

bearing WT mice, compared with that of ICB treatment (Fig. 5b and Supplementary Fig. 6c).

Consistent with the *Uba6*-null melanoma model, *Uba6*-null 4T1 tumours implanted in NSG mice showed a modest reduction in tumour volume and limited benefit in survival, whereas *Uba6*-null 4T1 cells did not show any growth disadvantage in vitro (Fig. 5c and Supplementary Fig. 6d, e). In WT mice, *Uba6*-null 4T1 tumours were completely rejected within two weeks (Fig. 5c). Notably, ICB or combination with inosine treatment did not exhibit further benefits (Fig. 5d and Supplementary Fig. 6f).

The dramatic biology of *Uba6*-null 4T1 and B16-GMCSF tumour-bearing mice was consistent with the relatively higher expression of UBA6 in these two tumour cell lines (Supplementary Fig. 6g). Unlike 4T1 and B16-GMCSF tumour cell models (Fig. 2e), inosine did not enhance T-cell-mediated tumour cell killing in inosine-pretreated MC38 tumour cell model (Supplementary Fig. 6h), which have the relatively lower expression of UBA6 protein (Supplementary Fig. 6g). Also, inosine did not significantly impact the expression of antigen presentation-related genes and inflammatory-related genes in MC38 cells (Supplementary Fig. 6i). However, inosine significantly increased the sensitivity of UBA6-overexpressed MC38 cells to T-cell-mediated tumour cell killing, compared to control MC38 cells (Supplementary Fig. 6j). These results suggest that tumoural UBA6 expression level determines the tumour immunogenicity and contributes to the direct effect of inosine on tumour cells, which indicate the potential application of UBA6 as a diagnostic or predictive biomarker for immunotherapy.

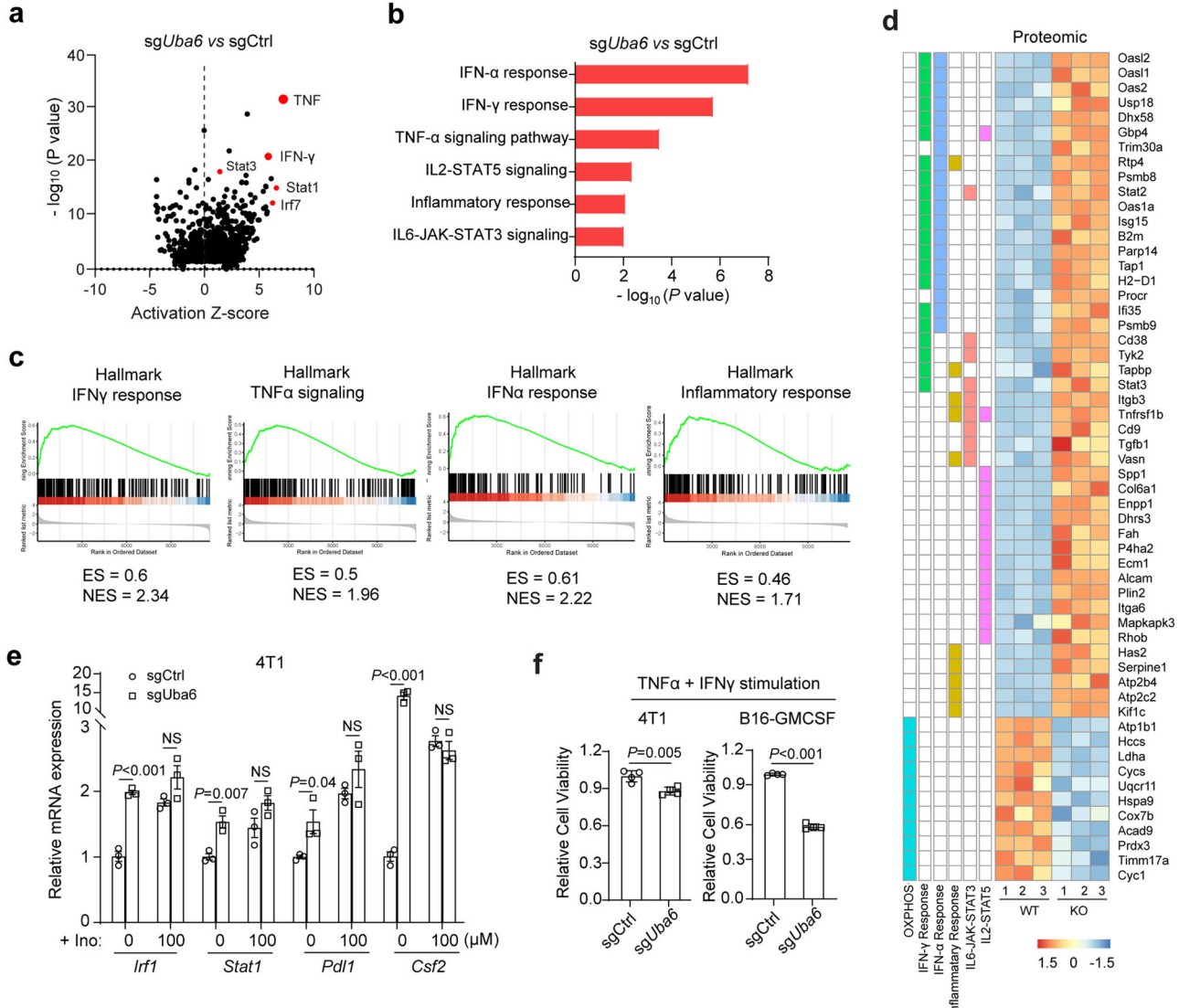

**Fig. 4 | Inosine and genetic inhibition of UBA6 stimulate tumour cell-intrinsic immune response signalling. a** Volcano plots of upstream regulator analysis of UBA6-dependent genes in 4T1 tumour cells by IPA. **b** Top-ranked GO terms in the transcriptome of *Uba6*-null 4T1 tumour cells. **c** The upregulated GSEA signatures in the *Uba6*-null 4T1 tumour. **d** Heat-map of proteins for differential signalling pathways (Red upregulated, blue downregulated). The expression level of these proteins in sgCtrl and sg*Uba6* 4T1 cells is measured by the whole proteomics (*n* = 3). **e** The relative mRNA expression of IFNγ response-related genes in sgCtrl and

sgUba6 4T1 tumour cells without or with inosine (100 μM) treatment for 48 h (*n* = 3). **f** Cell viability of *Uba6*-null and control 4T1 or B16-GMCSF tumour cells following stimulation with 10 ng/ml TNFα + 10 ng/ml IFNγ treatment for 48 h (*n* = 4). Data are presented as Mean ± s.e.m. Statistical significance was determined by two-sided Student's *t* test (**e, f**). The *P* values in **a** and **b** were based on log-rank tests. NS no significant. Raw RNA-seq data is available in the GEO database with accession number GSE210225. For the remaining data, source data are provided in a Source Data file.

## UBA6 expression predicts immunotherapy responses in clinical patients

Finally, we investigated the relationship between *UBA6* expression and immunotherapy response in cancer patients. *UBA6* was highly expressed in human tumours compared to normal tissues (Supplementary Fig. 7a) and low *UBA6* expression was associated with improved OS of patients in several tumour types (Supplementary Fig. 7b, c). Using the computational TIDE data sets[43], we found that a higher CTL level was associated with better survival in melanoma patients with a low expression of *UBA6*, but not a high expression (Fig. 6a). Moreover, this correlation was also obtained in cohorts with metastatic triple-negative breast cancer (TNBC), lung cancer, and melanoma with anti-PD1-resistance[44] (Supplementary Fig. 7d–f). This observation indicates the potential important function of *UBA6* in initiating immunotherapy resistance.

To directly evaluate the relationship between *UBA6* expression and ICB responses, we analysed the clinical dataset in a melanoma cohort treated with anti-CTLA4[45] and observed that *UBA6* expression was negatively predictive of the progression-free survival of ICB-treated patients (Fig. 6b). Similar trends were observed in additional independent cohorts of melanoma patients treated with anti-PD1[46] (Supplementary Fig. 8a). To further validate the relationship between tumoural UBA6 protein expression and immunotherapy responses in other cancer patients, we measured UAB6 protein expression pattern and CD8⁺ T cells infiltration in tumour tissues collected from a cohort of cancer patients (*n* = 22) enroled in a basket trial of anti-PD1 treatment, which included oesophageal cancer, lung cancer, stomach cancer, and colon cancer patients. We found markedly lower levels of tumoural UBA6 in cancer patients with partial response (PR), or stable disease (SD) compared to that in cancer patients with progressive

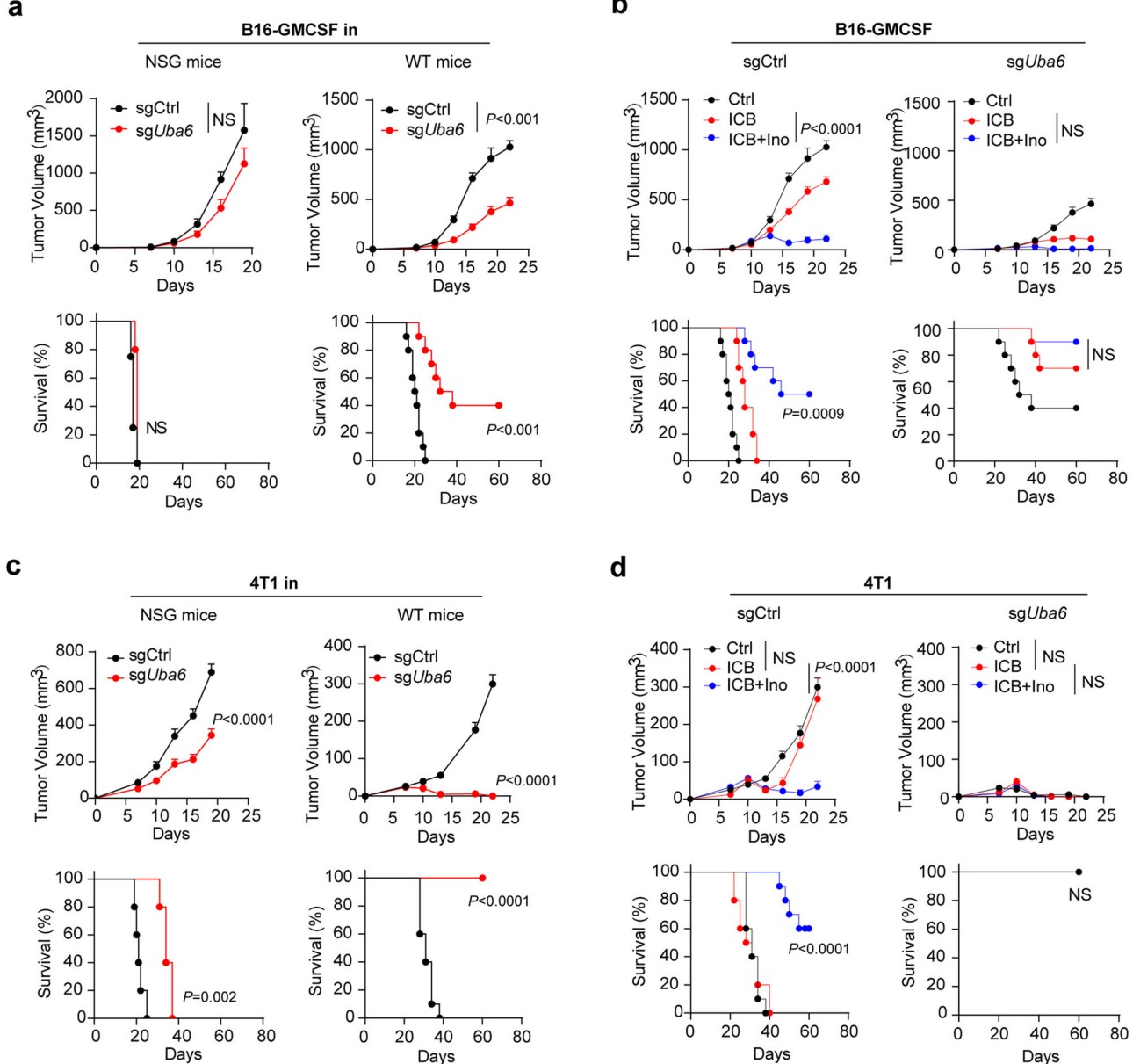

**Fig. 5 | UBA6 deletion substitutes the effect of inosine on antitumour immunity in vivo. a** Tumour volume and survival analysis of sgCtrl and sg*Uba6* B16-GMCSF tumours in NSG, wild-type (WT) mice (*n* = 8). **b** Tumour volume and survival analysis of sgCtrl and sg*Uba6* B16-GMCSF tumours in WT mice with Ctrl, ICB, or ICB + Ino treatment (*n* = 8). **c** Tumour volume and survival analysis of sgCtrl and sg*Uba6* 4T1 tumour-bearing NSG mice (left) or wild-type (WT) mice (right) (*n* = 10).

**d** Tumour volume and survival analysis of sgCtrl and sg*Uba6* 4T1 tumour-bearing WT mice with Ctrl, ICB, or ICB + Ino treatment (*n* = 10). Data are presented as mean ± SEM. Statistical significance was determined by ANOVA (tumour volume of **a**–**d**) or log-rank (Mantel–Cox) test (survival analysis of **a**–**d**). NS no significant. Source data are provided as a Source Data file.

disease (PD) (Fig. 6c–e and Supplementary Fig. 8b, c). We separated cancer patients into two groups based on their tumoural UBA6 level. The patients with low levels of UBA6 had a higher overall response rate (ORR) compared to patients with high UAB6 levels (ORR: 50% vs 17%) (Fig. 6d, e and Supplementary Fig. 8c). In addition, the UBA6 protein expression level on tumour cells was negatively correlated with CD8+ T cells infiltration in the TME (*R* = 0.64, *P* < 0.01, Supplementary Fig. 8d), indicating that tumoural UBA6-low expression is associated with a strong antitumour immune response, consistent with the finding in the mouse tumour models (Fig. 2f). Altogether, these data from immunotherapy-treated cancer patients support the clinical relevance of our experimental findings from mouse tumour models with *UBA6* knockout (Fig. 5), suggesting that UBA6 in tumour cells would be a useful target for immunotherapy.

## Discussion

Our results demonstrate that inosine overcomes tumour cell-intrinsic resistance to immunotherapy by inhibiting UBA6 in tumour cells to enhance tumour immunogenicity (Fig. 7). We identify UBA6 functions as a tumour-intrinsic checkpoint that limits antitumour immunity and implicate UBA6 as an attractive target for immunotherapy. Together with recent studies[25,47], our findings highlight the potential application of inosine in combination with ICB for cancer patients with high UBA6 expression.

The metabolic alterations in some cancer patients following treatment with ICB generate an immunosuppressive TME that orchestrates the resistance to immunotherapy[14]. Notably, the immunosuppressive TME is characterised by metabolic imbalance, such as nutrient shortage and abundant immunosuppressive metabolite

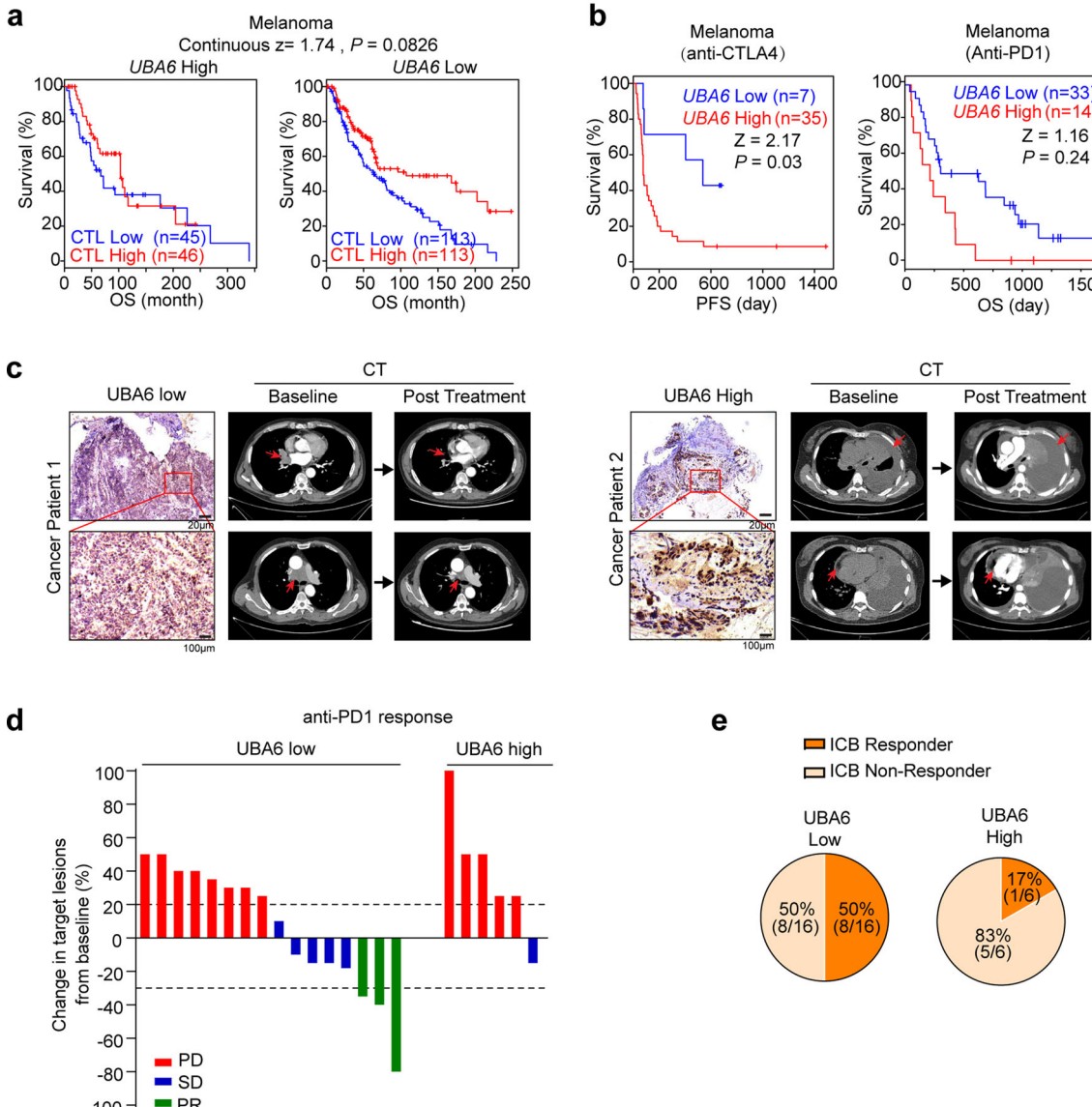

**Fig. 6 | UBA6 in tumour cells predicts patient outcomes to immunotherapy.**
**a** The association between CTL and OS of melanoma patients with distinct *UBA6* levels. **b** Kaplan–Meier plots of PFS of melanoma patients with anti-CTLA4 (*n* = 42) and OS of melanoma patients with anti-PD1 (*n* = 47) based on *UBA6* level. **c** Representative UBA6 protein staining tumour sections (top: 100×, bottom: 400×) (left) and CT scans (right) of lung cancer patients with anti-PD1 treatment. CT scans of tumours (top) and mediastinal lymph nodes (bottom) of patient 1 and left pleural effusion (top) and pericardial effusion (bottom) of patient 2 are highlighted by red arrows. Data were representative of three independent experiments (*n* = 3). **d** Waterfall plot depicting the responses to anti-PD1 treatment by the best change in the sum of target lesions, in comparison to baseline, in cancer patients with low UBA6 (*n* = 16) or high UBA expression (*n* = 6). Every bar represents one patient and the colours correspond to response to anti-PD1 treatment (*PR* partial response, *SD* stable disease, *PD* progressive disease). Dotted black lines indicate the response as described by RECIST1.1. **e** Pie charts of response fractions for each group of patients with UBA6-low and UBA6-high expression in tumour cells. Data are presented as Mean ± s.e.m. Statistical significance was determined by a log-rank test (**a**, **b**). Source data are provided as a Source Data file.

adenosine[48]. Inosine is a naturally occurring metabolite of adenosine and the circulating level of inosine is impacted by diet, genetics, and drugs[49,50]. Gut microbiota also contributes to inosine levels because it has been demonstrated that faecal microbiota transplantation or probiotics can reverse inosine depletion in vivo[25,41,51]. Moreover, inosine is synthesised and secreted by cancer cells[52]. But how immunotherapy alters the circulating level of inosine will be further explored.

Despite emerging evidence indicating that inosine has potent immunomodulatory effects[30,41], the mechanisms underlying the effect of inosine remain incompletely understood. Recent studies demonstrate that inosine improves immunotherapy response by being an alternative carbon source for CD8+ T-cell function under glucose restriction[47] or directing the differentiation of Th1 cells in an adenosine receptor $A_{2A}$-dependent manner[25]. Aside from the effect of inosine on T cells, we surprisingly identified that the increased tumour cell immunogenicity also contributed to the function of inosine for driving antitumour immunity and enhancing current immunotherapy. The complementary mechanisms of inosine on tumour cells in combination with ICB targeting T cells reasonably explain the superiority of combinational therapy in several tested mouse models. Thus, these findings in certain contexts indicate the pleiotropic effects of inosine on antitumour immunity by the complex and multiple action modes of the interactions between inosine and distinct components within TMEs.

Recently, it is recognised that beyond their roles as energy sources, metabolites serve as signals that trigger adaptive responses by functional interactions between metabolites and proteins[31]. Notably, our chemical proteomics indicated the specific binding of inosine to

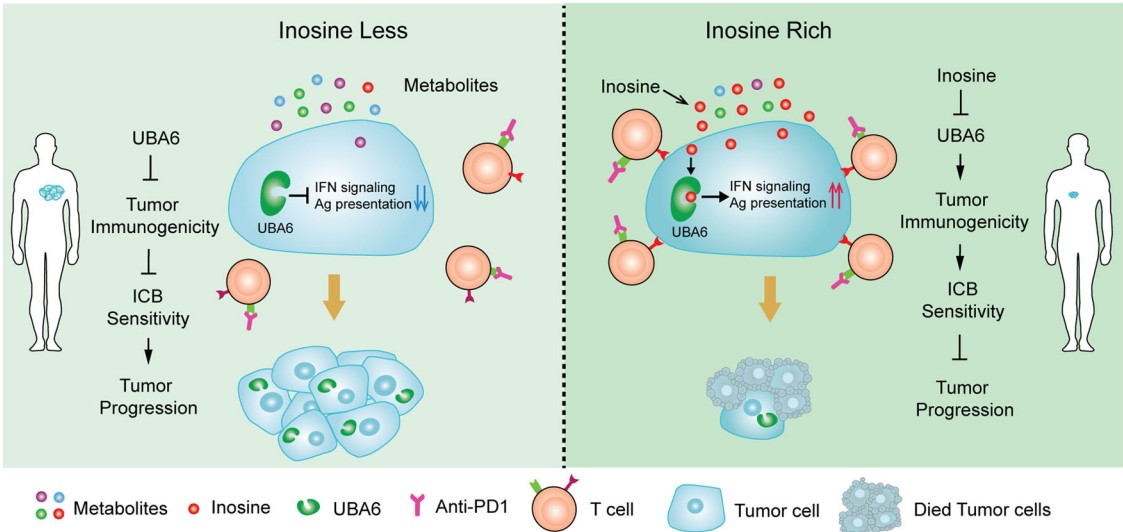

**Fig. 7 | A summary model linking inosine and immunotherapy responses.** The metabolic imbalance, especially inosine, is associated with immunotherapy responses in mice and humans. Inosine overcomes tumour-intrinsic resistance to immunotherapy by inhibiting UBA6 and increasing tumour immunogenicity in tumours with UBA6-high expression.

UBA6, and in vitro biochemical assay validated the inhibitory effect of inosine on UBA6 activity. UBA6 is a ubiquitin-activating enzyme that activates ubiquitin and ubiquitin-like protein, FAT10[35]. UBA6 plays an important role in embryogenesis and multiple pathogeneses of diseases including cancer progression and metastasis[36,37,53], however, the impact of UBA6 on tumour-intrinsic immunogenicity has been never addressed before. Interestingly, a recent report indicates the important role of ubiquitin-proteasome system (UPS) dysregulation in human cancer and underscores the potential therapeutic utility of targeting the UPS[54]. Despite abnormal expressions in UBA6 being found in several types of carcinomas, the function of UBA6 in anti-tumour immunity and immunotherapy is unclear. Here, a systematic series of genetic loss-of-function studies showed that loss of function of UBA6 in tumour cells led to tumour inflammation, and overcame resistance to ICB immunotherapy. These data indicate a critical role for UBA6 in the function of antitumour immunity and ICB therapy. However, the detailed molecular mechanism of how inosine modulates UBA6 and further details about the UBA6-dependent cell-intrinsic effects remain to be defined in cancer patients.

In addition, there are limitations inherent to our study with the modest numbers of cancer patients enroled to validate the predictive role of UBA6 for immunotherapy efficacy. It will be important to extend our findings to larger cohorts across tumour types. Due to intact samples missing, we did not validate and perform the inosine measurement or transcriptomics analysis in our cohort. Notably, adenosine could rapidly degrade into inosine (~10 s) and rapidly clear in plasma in vivo (in ~30 s)[55], whereas inosine has a much longer biological half-life (~15 h) than adenosine in vivo[56]. The short half-life of adenosine would impact inosine measurements in human plasma. Future studies attempting to further parse them are warranted. Despite these limitations, our study significantly expands our knowledge of metabolite inosine augments ICB efficacy by targeting tumoural UBA6 to enhance tumour immunogenicity. The findings of our proof-of-concept study not only provide molecular insight into how inosine triggers antitumour immunity but also suggest the application of inosine or targeting UBA6 for more effective immunotherapy.

## Methods
### Mice and cell lines
Female WT C57BL/6, BALB/c, and NOD-SCID IL2Rg[null] (NSG) mice (6–8 weeks old) were purchased from Shanghai Jie Si Jie Laboratory or Beijing Biocytogen and allowed to acclimatise for 1–2 weeks before experimentation. All animal experimental procedures were approved by the Institutional Animal Care and Use Committee of Shanghai General hospital affiliated with Shanghai Jiao Tong University School of Medicine (2019-A012-01).

The cell lines B16-F0, B16-F10, 4T1, and MC38 were obtained from the American Type Culture Collection (ATCC). B16-GMCSF cells were generated by retroviral-mediated gene transfer, following the previously described[57]. B16-GMCSF-OVA was ovalbumin (OVA)-transfected clone derived from B16-GMCSF cells which were transfected with the plasmid pCI-neo-mOVA (Cat. 25099, Addgene). Cells were cultured using RPMI-1640 (Corning) with 10% fetal bovine serum (Corning) and 1% Pen/Strep (GIBCO). Cells were incubated in an incubator maintained at 37 °C and 5% $CO_2$.

### Tumour challenge and treatment
For the B16 tumour challenges, $2 \times 10^5$ B16-F0 or B16-GMCSF tumour cells were resuspended in Hanks balanced salt solution (Gibco) and intradermally (i.d.) injected into the right flank of C57BL/6 J mice on day 0. For the 4T1 model, $2 \times 10^5$ 4T1 cells were orthotopically injected into the mammary fat pad (MFP) of BALB/c mice on day 0. For studies in immune-compromised mice, the *Uba6*-null or control 4T1 cells were done in the NSG mice. Treatments were given as single agents or in combinations with the indicated regimen for each drug. Inosine (Cat. 4060, Sigma-Aldrich) was administered by oral gavage once a day at 400 mg/kg. Control groups received vehicles (sterilised water). Treatment was initiated on day 4 and ended on day 21 post tumour implant. The combination of Rat monoclonal anti-CTLA4 antibody (100 µg per mouse, clone 9H10, Bio X Cell) and anti-PD1 antibody (200 µg per mouse, clone RPM1-14, Bio X Cell) (ICB) treatment were injected intraperitoneally (i.p.) on days 7, 10, 13 and 16 for the indicated tumour models. Rat IgG2a isotype control was used in control mice corresponding to the ICB treatment group. Each tumour was measured every 3 days with a caliper beginning on day 7 after the challenge until either the survival endpoint was reached, or no palpable tumour remained. Tumour volume was calculated using the formula: $(L \times W^2)/2$ and expressed as mm³. Mice that had no palpable tumours that could be measured on consecutive measurement days were considered complete regressions.

## T-cell-mediated cytotoxicity assays

CD8[+] T cells were isolated from the spleen of Balb/c or C57BL/6 mice using a CD8a[+] T-cell isolation kit (MiltenyiBiotec, Germany) according to the manufacturer's protocol. and then cultured in complete RPMI-1640 media (10% FBS, 20 mM HEPES, 1 mM sodium pyruvate, 0.05 mM 2-mercaptoethanol, 2 mM l-glutamine, and 50 U/ml of streptomycin and penicillin). Freshly isolated CD8[+] T cells were stimulated with anti-CD3/CD28 antibody (BioLegend, USA) to induce differentiation into an effector state. On day 3, recombinant mouse IL-2 (BioLegend, USA) was added to the culture at 20 ng/ml. For the generation of activated OT-1 T cells, splenocytes were harvested from the spleen of OT-1 transgenic mice and stimulated with 100 ng/mL of OVA peptide (SIINFEKL) for 24 h to expand CD8[+] OT-1 T cells. After washing to remove the peptide, cells were cultured in media for an additional 2 days before use in co-culture assays. 4T1 and B16-GMCSF cells were maintained in complete RPMI-1640 media. For the effect of inosine on T cells, isolated CD8[+] T cells were pretreated with a serial dilution of inosine for 48 h during T-cell activation. After washing, in vitro-activated CD8[+] T cells were co-cultured with indicated tumour cells at different effector-to-target ratios. For the effect of inosine on tumour cells, 4T1 cells or B16-GMCSF-OVA were seeded and pretreated with a serial dilution of inosine for 48 h, after washing and then co-cultured with activated CD8[+] T cells or OT-1 T cells respectively at a different effector to target ratios. Tumour cells were plated at equal density in all wells and activated CD8[+] T cells were added at target-to-effector 1:0, 1:2, 1:5 ratio (Target: Tumour cells; Effector: activated CD8[+] T cells). Cell viability is calculated as the quantification of the number of live cells and is also expressed as relative cell viability by calculating the FC of remaining alive target tumour cells following the incubation with T cells at the indicated inosine treatment compared to that in the untreated control. After a two or three-day co-culture with T cells, the number of viable tumour cells was counted using the automated cell counting system.

## Evaluation of toxicity of inosine in vivo

4T1 cells ($2 \times 10^5$) were implanted in BALB/c female mice. Mice were treated with vehicle (Ctrl), inosine, or a combination of inosine and ICB (anti-CTLA4 and anti-PD1 Abs) as described above. The body weight for all three groups of mice was measured every three days. On day 25, serum, liver, and lung were collected from mice with Ctrl, inosine, or ino+ICB treatment. The livers and lungs from different groups were fixed and processed and stained with hematoxylin and eosin. The serum ALT and AST levels from different groups were determined using assay kits, according to the manufacturer's instructions (Kehua Bio-Engineering, Shanghai, China).

## RNA-seq transcriptome analysis of tumour cells

Total RNA of *Uba6*-null or sgCtrl 4T1 cells was extracted from cell pellets and libraries prepared with the NEB Next Ultra Directional RNA Library Prep Kit for Illumina (New England Biolabs, USA) were sequenced on an Illumina NextSeq 500 instrument. Clean reads obtained by filtering the raw reads with Cutadapt (v 1.9.1) were aligned to the mouse reference genome (assembly GRCm38) using the HISAT2 v2.1.0. and subsequently assembled using Stringtie (v 1.3.3). Cuffdiff (v1.3.0) was applied to calculate Fragments Per Kilobase of exon per Million fragments mapped (FPKMs) for coding genes in each sample, and differentially expressed genes calling was applied using DESeq2 (v 1.30.1), in which significance was assessed by Benjamini-Hochberg False Discovery Rate (FDR) to account for multiple hypothesis testing. ClusterProfiler (v 3.18.1) was used to annotate genes with gene ontology (GO) terms and perform GSEA using the Hallmark gene signature collection from mSigDB[58]. Ingenuity Pathway Analysis (QIAGEN) was used for ingenuity upstream regulator analysis[59].

## Proteomics analysis of tumour cells

About $10^7$ of *Uba6*-null and sgCtrl 4T1 cells were suspended in a solution of 9.5 mol/L urea, 1% DTT, 40 ml/ml protease inhibitor cocktail, 0.2 mmol/L Na2VO3, and 1 mmol/L NaF. The mixture was centrifuged at $40,000 \times g$ at 15°C for 1 h and the supernatant was collected. The sequencing-grade trypsin was added to the supernatant containing about 1.5 mg of protein at an enzyme-to-protein ratio of 1:50 and incubated at 37 °C for 14 h. The peptides were desalted using a 1.3 ml C18 solid-phase extraction column (Sep-Pak® Cartridge) (Waters Corporation, Milford, USA) and analysed by two-dimensional (2D) strong cation-exchange/reversed-phase (RP) nano-scale liquid chromatography/mass spectrometry (2D-nanoLC/MS). Proteins and peptides were identified using a target-decoy approach with a reversed database and queried against the Mouse UniProt FASTA database. The quantification of peptides and proteins with "label-free quantification" (LFQ) was performed by MaxQuantv1.6.0.1.3.

## Identification of proteins interacting with inosine

Chemical proteomics by LiP-SMap approach was performed as in previous studies[31]. At first, 4T1 cells were lysed by bead-beating in PBS at 4 °C. After centrifugation at $16,000 \times g$ for 10 min at 4 °C, the supernatant was collected and aliquoted in equivalent volumes containing 100 μg of proteins each. To identify the proteins that interacted with inosine, 0.33 nmol/μg (total protein) of inosine was added to each aliquot and incubated at 25 °C for 10 min. Limited proteolysis was conducted by adding protein kinase K (Sangon Biotech, China) at a 1:100 enzyme/substrate ratio. The generated protein fragments were digested by trypsin with a 1:50 trypsin/substrate ratio to generate peptides for mass spectrometry analysis. Peptide fragments were analysed by Nano Acuity Ultra-High-Pressure liquid chromatography coupled with Thermo Q Exactive mass spectrometer (Thermo Fisher, USA). Proteins and peptides were identified using a target-decoy approach with a reversed database and queried against the Mouse UniProt FASTA database. The quantification of peptides and proteins with "label-free quantification" (LFQ) was performed by MaxQuant.

## Thioester assay

Recombinant human ubiquitin and FAT10 were purchased from Boston Biochem. Plasmids pEnter-UBA6 and pEnter-USE1 were used for the expression of Flag-UBA6 and Flag-USE1. The two plasmids were transfected into HEK293, respectively, using Lipofectamine 2000 reagent. Purification of Flag-UBA6 and Flag-USE1 was carried out using Anti-FLAGM2 Affinity Gel (Sigma) as described by the manufacturer's instructions. Flag-UBA6 (0.5 μM) and Flag-USE1 (0.5 μM) were incubated with vehicle, inosine, or TAK243 at room temperature for 15 min. Then, ubiquitin (5 μM) or FAT10 (2 μM) with ATP (250 μM) were added. The reaction mixtures were incubated at 37 °C for 30 min before 2× Lammli sample loading buffer was added to quench the reaction. The thioester detection was fractionated by SDS-PAGE under nonreducing conditions and immunoblotted with anti-UBA6 antibody (Proteintech, 1:1000) and anti-USE1 antibody (ABclonal, 1:1000).

## CRISPR screens in the 4T1 model

We designed a pooled sgRNA library targeting the 23 genes and 3 control genes (All sgRNA sequences are listed in Supplementary Data 6). It was cloned into LentiCRISPR V2 and delivered this pool to 4T1 cells. After transduction, 4T1 cells were selected in vitro for 7 days before transplantation into recipient mice. $2 \times 10^5$ pool-infected 4T1 cells were orthotopically injected into the MFP of WT and NSG mice on day 0. 4T1 tumours in NSG and WT mice were collected on day 12 and day 16, respectively. Genomic DNA was isolated using TIANamp Genomic DNA Kit (TIANGEN). PCR was then used to selectively amplify the sgRNA region and determination of sgRNA abundance was performed by Illumina sequencing as in the previous study[60].

## Generation of CRISPR-edited tumour cell lines

*Uba6* was deleted in Cas9-expressing 4T1 and B16-GMCSF mouse tumour cell line for validation experiments using a lentiviral delivery system (*lentiCRISPR v2*, Addgene) to express sgRNAs, and puromycin selection. For determining the knockout efficiency of the *Uba6* gene, western blotting was used to measure the protein expression of UBA6 in sgCtrl control and sg*Uba6* 4T1 or B16-GMCSF cells. The *Uba6*-null 4T1 or B16-GMCSF cells were selected for experiments.

## Metabolomic analysis in plasma

Plasma metabolites in tumour-free mice, B16-F0 tumour-bearing mice, and B16-F0 tumour-bearing mice with ICB treatment or Abx-treated mice were measured. A total of 244 metabolites in plasma were detected by ultra-high-performance liquid chromatography coupled with a tripleTOF 5600 plus mass spectrometer (Applied Biosystems, USA). The metabolomic data were analysed by pattern recognition analyses (principal component analysis and Heat-map).

## Antibiotic treatments

Six-week-old C57BL/6 J mice were treated with a cocktail of broad-spectrum antibiotics (1 g/L ampicillin, 1 g/L neomycin, 1 g/L metronidazole, and 0.5 g/L vancomycin) in drinking water for 3 weeks. The mice were allowed 3–4 days to recover before tumour implants. For measuring the levels of purine metabolites, the fresh faecal pellets, and plasma were collected on day 0 after 2 hours in collection cages with a paper liner. For evaluating the effect of ICB on tumour growth, IgG2a or a combination of anti-CTLA4 and anti-PD1 Abs (ICB) treated B16-F0 tumour-bearing mice as the indicated time points.

## The correlation analysis between survival and metabolites in human cancer patients

We reanalysed the public dataset[15] regarding the metabolic profiles of 743 RCC patients (Phase III trial: CheckMate 025, NCT01668784), among which 394 patients received nivolumab and 349 patients received everolimus. The OS in nivolumab or everolimus-treated RCC patients grouped by metabolite level (the upper half was as a high-level group; the lower half was as a low-level group defined by the median value of individual metabolites) was measured using Kaplan–Meier plot.

## Faecal microbial community analysis

Faecal DNA was extracted by using the DNeasyPowerSoil Kit (QIAGEN, Inc., Netherlands), according to the manufacturer's protocol. The sequencing of regions V3–V4 of the 16 S rRNA gene was performed using the Illumina MiSeq platform with MiSeq Reagent Kit v3 (Personal Biotechnology, Shanghai, China). The composition of the stool microbiota was analysed by QIIME-based microbiota analysis[61].

## Quantitative real-time PCR (qPCR)

Total RNA of indicated tumour cells was extracted by using RNA-prep pure Tissue Kit (Tiangen Biotech, China), according to the manufacturer's protocol. RNA (2 μg) was reverse transcribed using the PrimeScript™ RT reagent Kit with gDNA Eraser (Takara, China). Quantitative RT-PCR was performed using PowerUp™ SYBR™ Green Master Mix with QuantStudio 6 Flex System (Thermo Fisher, USA). Real-time PCR was run on the StepOnePlus system (Thermo Fisher) and data were analysed by StepOne Software v2.2.2. Relative mRNA expression was determined by the $^{\Delta\Delta}$Ct method and normalised to *Gapdh*. All qPCR primers used are listed in Supplementary Data 6.

## Western blot

ORF of human UBA6 and USE1 in pEnter, with C terminal Flag, were purchased from Vigene Biosciences (JiNan, China). Moreover, UBA6

with the UFD domain (residues 949-1052)deletion (UBA6$^{\Delta UFD}$) was generated by PCR. The amplified DNA fragment was cloned into pEnter. The human embryonic kidney cell line HEK293 was purchased from ATCC and was cultured in DMEM supplemented with 10% of FBS and 50 U/ml of penicillin/streptomycin. Cells were transfected using Lipofectamine 2000 reagent as described by the manufacturer's instructions. 500 μM of inosine or vehicle was added 24 h after transfection. At 48 h, cells were harvested and lysed in lysis buffer (50 mM Tris HCl, pH 7.4, 150 mM NaCl, 1 mM EDTA, 1% TRITON X-100). Cleared lysates were subjected to anti-FLAG immunoprecipitation using Anti-FLAGM2 Affinity Gel (Sigma, USA) overnight at 4 °C. Samples were washed three times with TBS. Proteins were separated on 8% or 12% Laemmli SDS gels and subjected to western blot analysis using an anti-UBA6 antibody (Proteintech, 1:1000), anti-USE1 antibody (ABclonal, 1:1000), and anti-β-Actin antibody (Sangon Biotech, 1:1000).

## UBA6/USE1 interaction assay in vitro

The proteins of Flag-UBA6 and Flag-USE1 were purified as described in Thioester Assay. Flag-UBA6 (0.5 μM) was incubated with vehicle or inosine (500 μM) at room temperature for 15 min. Then, Flag-USE1 (0.5 μM) was added and the reaction mixtures were incubated at 37 °C for 30 min. The solution was incubated with Anti-USE1 coupled to Protein A/G agarose beads (MedChemExpress) at 4 °C overnight (12 h). Beads were washed extensively and then eluted with sample loading buffer. The elution was heated at 100 °C for 10 minutes before western blot analysis using an anti-UBA6 antibody (Proteintech, 1:1000), and an anti-USE1 antibody (ABclonal, 1:1000).

## Prepare the single-cell suspension from 4T1 tumour tissues

4T1 cells ($2 \times 10^5$) were implanted in BALB/c female mice. Mice were treated with IgG2a (Ctrl) or a combination of inosine and ICB (anti-CTLA4 and anti-PD1 Abs) (Combo) as described above. On day 13, tumours were harvested and minced with scissors before incubation with collagenase A (2 mg/ml, Roche) and DNase I (50 μg/ml, Roche) in RPMI-1640 completed medium (10%FBS, 1% P/S) for 30 min at 37 °C. Tumour samples were homogenised by repeated pipetting and filtered through a 70 μm nylon filter (BD Biosciences) in FACS staining buffer (PBS/0.5% albumin) to generate single-cell suspensions. After red blood cell (RBC) lysis (RBC Lysing Buffer, Biolegend), all samples were washed and resuspended in FACS staining buffer. The cell viability for each sample should exceed 80% determined with trypan blue staining. An appropriate volume of cell suspension containing ~20,000 cells for each sample was used for further scRNA-seq or flow cytometry.

## Analysis of tumour-infiltrating immune cells by scRNA-seq

Tumour-infiltrating immune cells from 4T1 tumour-bearing mice with IgG2a (Ctrl, *n* = 2) or a combination of inosine and ICB (anti-CTLA4 and anti-PD1 Abs, *n* = 2) (Combo) treatment were enriched using CD45$^+$ MicroBeads kit (MiltenyiBiotec, Germany). Two biological replicates in the vehicle and inosine+ICB groups were performed. The sorted CD45$^+$ cells were suspended at a $1 \times 10^6$ cells/ml concentration in FACS buffer and the viability was higher than 80%. The single-cell RNA-seq was performed as described[62]. Briefly, cells were counted and loaded into the 10x Genomics device to generate single-cell Gel Beads-in-Emulsion (GEMs). After reverse transcription, barcoded cDNAs were purified, amplified, end-repaired, and ligated with Illumina adaptors to generate a single multiplexed library according to the manufacturer's protocol (10x Genomics). All resulting libraries were sequenced on the Illumina Novaseq 6000 platform (Illumina, USA), aiming at more than 50,000 reads per cell.

Preliminary sequencing results were de-multiplexed the cellular barcodes, aligned reads to the transcriptome GRCm38 (mouse), barcode counting, and unique molecular identifier (UMI) counting by the Cell Ranger v2.1.1 pipeline. For each sample, genes with detected expression in at least two cells were included and cells with less than

200 expressed genes and 200 UMI were excluded, and cells with greater than 10% of transcripts derived from mitochondrial genes or greater than 10% of transcripts derived from red cell genes were removed. To avoid low-quality cells, empty droplets, or multiplets, we further filtered cells based on the number of unique genes detected in each cell, which is capped in the range from 2.5th to 97.5th percentile. As a result, mean and dispersion values were calculated for each gene across the remaining 16,199 cells (Ctrl group) with a median of 1283 detected genes per cell and 9842 cells (Combo group) with a median of 1126 detected genes per cell, and variably expressed genes were selected for principal component analysis (PCA). The top 30 dimensions resulting from the PCA were used for the t-Distributed Stochastic Neighbour Embedding (tSNE). Then, tSNE was performed using default parameters for visualisation in two dimensions. All CD45$^+$ immune cells were clustered as described[63]. Unsupervised clustering using a shared nearest neighbour modularity optimisation-based algorithm identified 32 distinct clusters. 14 major clusters were identified by mapping canonical marker genes (All marker genes are listed in Supplementary Table 1) in the two-dimensional tSNE map.

## Flow cytometry assay

For flow cytometry analysis of in vivo experiments, tumour single cells were isolated from mouse 4T1 tumours as described above and pre-incubated (15 min, 4 °C) with an anti-CD16/32 monoclonal antibody (clone 93, Biolegend) to block nonspecific binding and then stained (30 min, 4 °C) with appropriate dilutions of various combinations of the following fluorochrome-conjugated antibodies: anti-CD45-AF 700 (clone 30-F11), anti-CD11b-PE (M1/70.15), anti-F4/80-APC (clone BM8), anti-MHC Class II-FITC (clone M5/114.15.2), anti-CD206-PE (clone 19.2), anti-CD8-Percp-Cy5.5 (clone 53-6.7), anti-CD4-PE (clone RM4-5) antibodies, all purchased from Biolegend or ThermoFisher. For tetramer staining in the 4T1 model, tumour antigen-specific CD8$^+$ T cells were detected with PE-conjugated H-2L$^d$ tetramer to peptide SPSYVYHQF (MuLV env gp70, 423–431) was purchased from MBL International. Antibodies were used at 5 µg/ml, and tetramer staining was performed in FACS buffer for 20 min at room temperature and followed by surface staining on ice for 20 min. Dead cells and doublets were excluded based on forward and side scatters and Fixable Viability Dye eFluor 506 (Thermo Fisher, USA).

For in vitro analysis of the effect of inosine on MHC Class I antigen expression, B16-GMCSF or 4T1 cells were seeded and treated with a serial dilution of inosine for 48 h. Cells were non-enzymatically detached from the wells, washed with FACS staining buffer, and then incubated with FITC-conjugated anti-mouse H-2K$^d$ antibody (clone SF1-1.1, BioLegend) for 30 min on ice. After washing, cells were resuspended in FACS staining buffer, and then >2000 cells were analysed by flow cytometry. Acquisition and analysis were performed on Canto II Flow Cytometer using BD FACSDiva software (BD Biosciences, USA) and all analyses were performed with FlowJo software v10 (BD).

## Tumour cells viability and apoptosis assays

For the effect of inosine on tumour cell growth in vitro assay, 4T1 or B16-GMCSF cells were seeded in 96-well plates (1000 cells per well) and allowed to seed for 24 h, after which they were treated with inosine. For in vitro cytokine stimulations and growth inhibition assays, sgCtrl or UBA6-*null* 4T1 or B16-GMCSF tumour cells were plated in media containing the indicated combinations of cytokines: 10 ng/ml IFNγ (PeproTech, USA), 10 ng/ml TNFα (PreproTech, USA), or 10 ng/ml IFNγ + 10 ng/ml TNFα. Treatment was given only once at the beginning, after the seeding of cells. Subsequently, every 24 h, MTT reagent (Sigma, USA) was added to the cell culture media for 3 h at 37 °C. The supernatant was then discarded and lysed with DMSO to dissolve the formazan product. Absorbance was measured by a spectrophotometric plate reader.

For flow cytometry analysis of apoptosis, 4T1 or B16-GMCSF cells were treated with inosine for 48 h, and following trypsinization and washes in FACS staining buffer, tumour cells were stained for 20 min on ice using the manufacturer's recommended concentrations of Annexin-V PE and 7-AAD from the PE Annexin-V Apoptosis Detection Kit 1 (BD Pharmingen, USA) according to the manufacturer's instructions. The staining of cell surface markers was then analysed using the Canto II flow cytometry system (BD Biosciences, USA). The analysis was carried out using FlowJo software.

## Integrative gene knockout screening platform and survival analysis based on TIDE

We collected cancer data sets with both patient survival durations and tumour gene expression profiles from The Tumour Immune Dysfunction and Exclusion (TIDE) website and tools[32,43]. Candidate genes were plotted based on mean log2 fold change (logFC) of gRNA counts compared to control selection and normalised z-score generated using the pheatmap R package and presented as the expression level of the individual gene was standardised to zero mean and one standard deviation. The normalised logFC and Z score in CRISPR screens help identify regulators/genes whose knockout can mediate the efficacy of lymphocyte-mediated tumour killing in cancer models. Higher logFC and Z-score mean that knockout of gene resistant to lymphocyte-mediated tumour cell killing, contrast, lower logFC, and Z score mean that knockout of gene mediates the enhancement to lymphocyte-mediated tumour cell killing. Kaplan–Meier plots of OS or disease-free survival (DFS) of cancer patients treated with ICT who had high UBA6 vs. low UBA6 in the tumours as respectively defined by the median expression levels were accessed by the TIDE programme. To test the association between UBA6 gene expression level and patient survival, Kaplan–Meier survival analysis was performed using the programme described in the Gene Expression Profiling Interactive Analysis (GEPIA2)[64].

## Human tissue samples

A total of 22 human tissue samples from patients with cancers of oesophageal (n = 7), stomach (n = 4), lung (n = 3), colon (n = 3), and others (n = 5) (gender: 12 males and 10 females; mean ± SD age, 62.4 ± 8.6 years; median age, 63 years; range, 39–77 years) were collected pre-anti-PD1 treatment (patients were treated with Sintilimab, Innovent Biologics) at the Beijing Friendship Hospital, Capital Medical University. Disease assessments were performed with the use of computed tomography (CT) or magnetic resonance imaging at baseline, every 8 weeks until disease progression or discontinuation of treatment. The clinical objective response was determined as the investigator-assessed best response based on immune-related response evaluation criteria in solid tumours (irRECIST)[65] using unidimensional measurements (CR: complete response, PR: partial response, SD: stable disease, PD: progressive disease). The best percentage change in the sum of the diameters for the selected target lesion was defined byRECIST. Prior to participation, written informed consent was obtained from each participant and/or their legal representative, as appropriate. All studies were performed in accordance with the Declaration of Helsinki and we complied with all relevant ethical regulations. The use of pathological specimens, as well as the review of all pertinent patient records, were approved by the Bioethics Committee of Beijing Friendship Hospital, Capital Medical University (project approval number 2017-P2-141-02).

## Histological evaluation of UBA6 expression

Standard immunohistochemical (IHC) assays were performed for UBA6 evaluation as described previously[66]. In brief, tumours were harvested before immunotherapy and fixed in 10% neutral-buffered formalin. After deparaffinization and rehydration, 4 µm tissue sections were subjected to heat-induced epitope retrieval. Slides were

processed with the VECTASTAIN Elite ABC HRP Kit and DAB Substrate Kit (Vector Laboratories). Slides were then incubated with anti-UBA6 antibody (Proteintech, 1:1500). Five visual fields from different areas of each slide were independently evaluated by two pathologists who were blinded to the group allocation during the staining and when assessing the outcomes. Necrotic areas in the tumours were excluded from the evaluation. IHC intensity scores of UBA6 were ranked into four groups: negative (−), positive-low (+), positive-medium (++), and positive-high (+++). In the IHC scoring of patient samples, the score "low" corresponded from negative (−) to positive-low (+), while the score "high" corresponded to the range from ++ to +++. To stain for CD8, slices were incubated with an anti-CD8 antibody (Cell signalling #85336 S, 1:200). CD8-positive cells were analysed using the Image J cell counter. The average infiltration of CD8$^+$ cells and average expression of UAB6 in the tumour tissues were assessed.

### Statistics and reproducibility

Statistical tests employed with the number of replicates and independent experiments are provided in the figure legends or text. Unless mentioned otherwise, all graphs with error bars are presented as mean ± s.e.m of at least three independent experiments. GraphPad Prism (v.8) is used for basic statistical analysis and plotting. Statistical significance is determined by one-way analysis of variance (ANOVA) with Tukey and Dunnett's posttests and two-way ANOVA with a Bonferroni test for multiple comparisons, or an unpaired Student's $t$ test for pair-wised comparison. Multiple hypothesis testing corrections were applied where multiple hypotheses were tested and are indicated using FDR. Kaplan–Meier survival curves are graphed and analysed using the log-rank test for multiple comparisons. $P$ value < 0.05 was indicated as statistically significant.

### Reporting summary

Further information on research design is available in the Nature Research Reporting Summary linked to this article.

## Data availability

Raw RNA-seq data are available in the GEO database with accession number GSE210225. The remaining data are available within the manuscript, supplementary information, or source data file. Source data are provided in this paper. Source data are provided with this paper.

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

## Acknowledgements

We thank Dr. J. Marc Rhoads for his helpful comments and Genergy Biotech (Shanghai) team for technical assistance. These studies were supported by The National Natural Science Foundation of China (81974256, B.H.; 81630102, Y.G.; 81772525, X.Y.; 81402148, L.J.; 81871274, D.X.), The National Key Research and Development Programme of China (2019YFC1604900, Y.G.), and Shenzhen High-level Hospital Construction Fund.

## Author contributions

B.H. and X.Y. conceived the project, guided the experimental design, and wrote the manuscript. Y.G., D.Y., and D.X. guided the experimental design and help to write the manuscript. Lei Zhang, L.J., L.Y., Q.L., X.T., J.H., C.W., Ling Zeng, Y.W., Jing Li, X.J., X.G., D.W., A.J.S., Q.G., Jikun Li, and Y.Y. performed all the experiments and data analysis. All authors contributed to the editing of the manuscript.

## Competing interests

The authors declare that they have no competing interests.

## Additional information

[1]Department of Gastroenterology, Shanghai General Hospital, Shanghai JiaoTong University School of Medicine, Shanghai 200080, China. [2]Shanghai Key Laboratory of Pancreatic Disease, Shanghai Jiao Tong University School of Medicine, Shanghai 201620, China. [3]Department of Gynecology and Obstetrics, Xinhua Hospital, Shanghai Jiao Tong University School of Medicine, Shanghai 200092, China. [4]Department of Surgery, Shanghai General Hospital, Shanghai Jiao Tong University School of Medicine, Shanghai 201620, China. [5]Department of Oncology, Beijing Friendship Hospital, Capital Medical University, Beijing 100050, China. [6]Department of Bioinformatics & Computational Biology, The University of Texas MD Anderson Cancer Center, Houston, TX 77030, USA. [7]Biotree Institute of Health, Shanghai 201800, China. [8]Department of Laboratory Animal Center, Shanghai General Hospital, Shanghai Jiao Tong University School of Medicine, Shanghai 200080, China. [9]Department of Clinical Laboratory Science, The Affiliated Hospital of Qingdao University, Qingdao, Shandong 266003, China. [10]Department of Biostatistics, UNC Gillings School of Global Public Health, University of North Carolina at Chapel Hill, Chapel Hill, NC 27599-7455, USA. [11]Department of Periodontology, School of Dentistry, University of North Carolina at Chapel Hill, Chapel Hill, NC 27599-7455, USA. [12]Hudson Institute of Medical Research, 27-31 Wright Street, Clayton, VIC 3168, Australia. [13]Department of Molecular and Translational Science, Monash University, Melbourne, VIC 3168, Australia. [14]Diamantina Institute, Faculty of Medicine, The University of Queensland, QLD 4102, Australia. [15]Department of Laboratory Medicine, Ruijin Hospital, Shanghai Jiao Tong University School of Medicine, Shanghai 200025, China. [16]Department of Pharmaceutical Sciences, Beijing Institute of Radiation Medicine, Beijing 100850, China. [17]Institute of Chinese Materia Medica, The Fourth Clinical Medical College, Guangzhou University of Chinese Medicine, Shenzhen, Guangdong 518033, China. [18]These authors contributed equally: Lei Zhang, Li Jiang, Liang Yu, Qin Li. ✉e-mail: gaoyue@bmi.ac.cn; yuanxiangliang@gmail.com; baokun.he@shgh.cn

