## [Peer Review File · Nature Communications]

Inhibiting tumour cell UBA6 expression by inosine augments tumour immunogenicity and response to immune checkpoint inhibitionREVIEWER COMMENTS

Reviewer #1 (Remarks to the Author):

Authors precluded that metabolite inosine was associated with ICB sensitivity in mice and humans, and overcame ICB resistance in several 43 mouse tumor models They tried to showed inosine directly bound and inhibited ubiquitin-activating enzyme UBA6. Tumor UBA6 loss augmented 46 tumor immunogenicity and substituted the synergistic effect of inosine in combination with ICB and tumor UBA6 expression negatively correlated with ICB response in cancer patients.

Although the findings are interesting, the data, particularly derived from the clinical samples are not convincing to support authors' conclusion.

Reviewer #2 (Remarks to the Author):

Zhang et al., have discovered that iosine increases viral associated genes in tumors and sensitizes cells to immune check point blockage. They have exciting data from mouse models and patient data sets. Their comparisons using immune competent and immune deficient mouse models are particularly exciting. They also discover an important role for Uba6 in ICB response.

I am very enthusiastic about this work. It has important mechanistic insights and strong therapeutic potential.

I have the following comments:

- 1) The authors note a decrease in iosine in tumor bearing mice treated with ICB. Is the presence of the tumor needed to the alter plasma levels of iosine. In other words, is a similar decrease in iosine seen in mice without tumors treated with ICB?
- 2) The authors find that patients treated with ICB who have low levels of iosine after treatment have improved response, but then show that supplementing mice with iosine increases response and increases sensitivity to ICB. Can the authors speculate on this discordance?
- 3) The authors use a mass spec study to support the binding of iosine to Uba6. Can they confirm direct binding in a cell free assay?
- 4) Can the authors comment on the toxicity of iosine in their mouse models
- 5) The introduction is relatively short and could be expanded.
- 6) There are multiple grammatical errors that make the paper difficult to follow in places

Reviewer #3 (Remarks to the Author):

The report "Inhibiting tumor cell-intrinsic UBA6 by inosine augments tumor immunogenicity" Zhang et al. touches on a relevant topic in anti-tumor immunity. They show that inosine sensitizes different tumor cells to T cell mediated killing in vitro and in vivo, particularly when combined with immune checkpoint blockade therapy . This effect could be inhibited using broad spectrum antibiotics (orally), indicating that the effect may depend upon the microbiome. They further reveal that inosine induces this phenotype through UBA6. Complementing the in vitro and in vivo murine studies they show that inosine is increased in the plasma of cancer patients responding to immune checkpoint blockade therapy. Lastly they also show that low UBA6 expression correlates with improved cancer patients survival and predicts the outcome of immune checkpoint blockade therapy.

Overall the study by Zhang et al. seems well performed and is relevant, particularly due to their dual approach (murine functional/mechanistic studies and validation in human samples/cohorts). However

several key points need to be addressed before publication.

Major concerns:

- 1) The cancer cell direct effect of inosine is very intriguing, however the precise circumstances are unclear considering previous reports in the field. Wang, et al. *Nat Metab.* 2020 and Mager et al. *Science* 2020 have shown a direct effect of inosine on T cells for anti-tumor immunity. However, this report did not see an effect when immune cells were treated with inosine (Fig. 2a). How do the authors explain that? Could it be due to insufficient activation of the T cells (only antigen stimulation without concurrent co-stimulation i.e. could anti-CD28 treatment change their findings in Fig. 2a). Moreover, Mager et al. *Science* 2020 describes inosine-tumor cell sensitization experiments similar to the experiments performed in this report and did not see an enhanced tumor cell death mediated by T cells. Is the direct effect of inosine on cancer cells dependent on the cell line? Does it only work on cell lines with no/low MHC1 expression that is enhanced after inosine treatment, but not on cell lines with high baseline MHC expression?
- 2) The authors nicely demonstrate that inosine mediates its function through altered UBA6- FAT10-USE1 interactions. However, it is unclear how inosine does so. Does inosine change UBA6- FAT10-USE1 interactions through inducing upstream signalling and its consequences (i.e. through A1, A2A, A2B and A3 receptors) or through equilibrative nucleoside transporters (i.e. ENT1 and ENT2) and consequent changes in the cytoplasm. Could the authors block the effect of inosine with adenosine receptor inhibitors or inosine transport inhibitors (or genetic tools)?
- 3) The gut microbiome and specific bacteria thereof have been shown to improve immune checkpoint blockade therapy (also through the production of inosine). The authors' own data using broad spectrum antibiotics (sFig. 1d-f) are in line with these observations. It would be important to at least profile the microbiome of the mice used in this study and evaluate whether certain bacterial taxa are enriched in mice treated with inosine and checkpoint blockade therapy compared to the other groups.
- 4) The authors rely on published datasets to some extent. (i.e. Fig. 1 and b, Fig. 3d, Fig. 6a and b, sFig. 1 g and h, sFig. 3 b and c, sFig. a-f). While the benefit of these datasets is obvious, there are certainly some drawbacks. First example: Using plasma samples from cancer patients to detect inosine: While inosine is rather stable (half-life of 15 hours), adenosine- the precursor of inosine- is not (half-life of 10 seconds). Thus without tight control of the sampling procedure/duration and more importantly adenosine stabilization, the levels of inosine could drastically be altered. Second example: the proteomics approach to identify inosine modulated proteins (Fig. 3a) identified 23 candidates. This was done in 4T1 breast cancer cells. UBA6 was then identified from these 23 candidates using a published database (Pan et al. *Science*. 2018), which relies however on B16F10 cells, a melanoma cell line. It is doubtful whether results in B16F10 can or should be used to draw conclusions from results obtained from 4T1 cells. While it is clear that the authors cannot repeat large human cohorts for their inosine measurements or transcriptomics analysis, the risks/limitations should certainly be discussed. However, the screen to identify UBA6 should be repeated by the authors with the same model system (4T1) in a mini-screen targeting their 23 candidates.

Minor concerns:

- 1) Gene nomenclature is not always consistent. This should be fixed. i.e. sFig. 6 legend (Uba6 vs UBA6)
- 2) The authors show that UBA6 is downregulated across all cancer types and that low UBA6 expression is favourable also in BRCA (absence of immune checkpoint blockade therapy) (sFig6). However, other reports have associated low UBA6 with cancer progression. i.e. Liu et al. *Oncotarget*. 2017 showed that UBA6 suppresses EMT and cancer invasion. The role of UBA6 in cancer is still unclear and should be discussed more cautiously.
- 3) The main results of the paper could be presented in more detail in the text to guide and help the reader a bit more through the key findings.

Name of journal: Nature Communications

Manuscript NO.: NCOMMS-21-22862

Manuscript Type: Research Article

Zhang L., et al. Inhibiting tumor cell-intrinsic UBA6 by inosine augments tumor immunogenicity.

Point-by-point response:

We sincerely thank the editorial team and the reviewers for spending their precious time with our manuscript and for their insightful comments and suggestions to further improve our manuscript. Following the reviewers' suggestions and the editorial guidance, we have performed additional experiments and revised the manuscript accordingly to address all concerns of the reviewers. The changes and additions in this revised manuscript were highlighted in yellow complies with Nature Communications requirements. Please see the point-by-point response to the reviewers' comments below following each reviewer's questions or comments in blue font.

Reviewer #1 (Remarks to the Author):

Authors precluded that metabolite inosine was associated with ICB sensitivity in mice and humans, and overcame ICB resistance in several mouse tumor models They tried to showed inosine directly bound and inhibited ubiquitin-activating enzyme UBA6. Tumor UBA6 loss augmented tumor immunogenicity and substituted the synergistic effect of inosine in combination with ICB and tumor UBA6 expression negatively correlated with ICB response in cancer patients.

Although the findings are interesting, the data, particularly derived from the clinical samples are not convincing to support authors' conclusion.

Response:

Thank the Reviewer for your comment, we appreciate that you find our work are interesting. As the Reviewer mentioned, our work touches on an important biological finding and dissects the mechanism of metabolite inosine which augments immunotherapy sensitivity by directly binding and inhibiting UBA6 activity in tumor cells. Moreover, as highlighted by other reviewers, the major significance of our study is that our work used a dual approach (murine functional /mechanistic studies and validation in human samples/cohorts) to strengthen our conclusion (refer to Reviewer 3' comment), which has important mechanistic insights and strong therapeutic

potential (refer to Reviewer 2' comment).

With regards to the Reviewer's comment regarding the weakness of data derived from clinical samples, we actually utilized our clinical patient cohort, along with several public patient datasets, to validate the key mechanistic findings which have been identified in several mouse tumor models. These results strongly support our conclusions and empower the clinical impact of our findings. Here, we would like to highlight our data generated from clinical evidence. Firstly, to validate the association of metabolite inosine with ICB efficacy, we analyzed the metabolic profiling of renal cell carcinoma (RCC) patients (Phase III trial: NCT01668784). The data showed that inosine was significantly associated with ICB response in humans (Figure 1a, b; and Supplementary Fig. 2g), consistent with the results in mice models (Supplementary Fig. 1b, c). Secondly, since UBA6 was identified as the direct target of inosine to impact the tumor immunogenicity, we analyzed the TIDE datasets and found the negative correlation of UBA6 with cytolytic CD8⁺ T cells activity (Fig. 6a; Supplementary Fig. 7d-f) and ICB efficacy in several independent cohorts of melanoma patients treated with ICB (Fig. 6b, Supplementary Fig. 8a). Finally, we directly examined whether tumoral UBA6 protein levels are associated with patients' response to ICB immunotherapy in a basket trial of anti-PD1 treatment. We found markedly that UBA6 expression in cancer patients who had clinical benefit from anti-PD1 based immunotherapy was lower than that in patients who did not respond to anti-PD1 treatment (Fig. 6c-d and Supplementary Fig. 8b-c).

To further strengthen our conclusion with clinical data, we performed additional experiments to analyze tumoral UBA6 and ICB (anti-PD1) efficacy based on a new analysis of best percentage change in the sum of the diameters for the selected target lesion as defined by Response Evaluation Criteria in Solid Tumors (RECIST version 1.1) on minimum 1 computed tomographic scans before treatment and 1 computed tomographic scan during treatment (Figure 6d). In addition, we measured UAB6 protein expression pattern and simultaneously CD8⁺ T cells infiltration in tumor tissues collected from a cohort of cancer patients (n=22) enrolled in a basket trial of anti-PD-1 treatment, which included esophageal cancer, lung cancer, stomach cancer, and colon cancer patients. We found a markedly lower level of tumoral UBA6 in cancer patients with partial response (PR) or stable disease (SD) compared to that in cancer patients with progressive disease (PD) (Fig. 6d, e and Supplementary Fig. 8c). We separated cancer patients into two groups based on their tumoral UBA6 level, patients with low levels of UBA6 had a higher overall response rate (ORR) compared to patients with high UAB6 levels (ORR: 50% vs 17%) (Fig. 6d, e and

Supplementary Fig. 8c). Notably, there were no significant differences in age, gender, and tumor stage among the two groups. Moreover, the UBA6 protein expression level in tumor cells is negatively correlated with CD8⁺ T cells infiltration in the tumor microenvironment (R=0.64, $P<0.01$, Supplementary Fig. 8d), indicating that tumoral UBA6 low expression is associated with a strong antitumor immune response, consistent with the finding in the mouse 4T1 tumor model (Fig. 5a, c). These data from immunotherapy-treated cancer patients support the clinical relevance of our experimental findings from mouse tumor models with UBA6 knockout (Figures 5), suggesting that UBA6 expression in tumor cells was predictive for ICB responses in cancer patients. Collectively, our proof-of-concept study brings forward a novel conceptual advancement in understanding the accurate mechanisms of inosine driving anti-tumor immunity by targeting UBA6 in tumors and discovers UBA6 as a new appealing target for immunotherapy.

We agree with the reviewer that there are limitations inherent to our study with the modest numbers of cancer patients enrolled to validate the predictive role of UBA6 for immunotherapy efficacy. It will be important to extend our findings to larger cohorts across tumor types. Additionally, due to intact samples missing, we did not validate and perform the inosine measurement or transcriptomics analysis in our cohort. Future studies attempting to further parse them are warranted. Despite these limitations, our study significantly expands our knowledge of metabolite inosine augments ICB efficacy by targeting tumoral UBA6 to enhance tumor immunogenicity. Furthermore, based on these promising findings in mouse models and strong clinically relevant data, we are encouraged to be planning a biomarker-guided precision clinical trial to test inosine administration for enhancing ICB response in cancer patients with high UBA6 expression. We pointed out these limitations in the Discussion section (Line 384-388).

Reviewer #2 (Remarks to the Author):

Zhang et al., have discovered that inosine increases viral associated genes in tumors and sensitizes cells to immune check point blockage. They have exciting data from mouse models and patient data sets. Their comparisons using immune competent and immune deficient mouse models are particularly exciting. They also discover an important role for Uba6 in ICB response. I am very enthusiastic about this work. It has important mechanistic insights and strong therapeutic potential.

We thank the Reviewer for the positive assessment of our work.

I have the following comments:

1) The authors note a decrease in inosine in tumor bearing mice treated with ICB. Is the presence of the tumor needed to alter plasma levels of inosine? In other words, is a similar decrease in inosine seen in mice without tumors treated with ICB?

Response:

We thank the Reviewer for bringing up this important point. In this study, we found a decreased plasma level of inosine in tumor-bearing mice treated with ICB (Supplementary Fig. 1b, c). Following the reviewer's suggestion, we measured the plasma levels of inosine in tumor-bearing mice and tumor-free mice. Our results showed that the plasma levels of inosine in tumor-bearing mice were higher than that in tumor-free mice (Supplementary Fig. 2f). Altogether, these findings demonstrate that the decrease of plasma inosine levels in tumor-bearing mice treated with ICB would be due to ICB treatment, but not due to the presence of the tumor.

2) The authors find that patients treated with ICB who have low levels of inosine after treatment have improved response, but then show that supplementing mice with inosine increases response and increases sensitivity to ICB. Can the authors speculate on this discordance?

Response:

We thank the Reviewer for pointing out this important question. In this study, our results showed that the inosine levels were significantly decreased in the plasma of tumor-bearing mice and humans after ICB treatment (Supplementary Fig. 1c, 2g). From the clinical data, the higher levels of inosine before being treated with nivolumab (Anti-PD1) were significantly associated with better overall survival (OS) of cancer patients (Fig. 1b). Indeed, ICB alone significantly reduced tumor growth in the B16-F0 model (Supplementary Fig. 1a). Notably, the new data indicated that the plasma levels of inosine in B16-F0 tumor-bearing mice were higher than that in tumor-free mice (Supplementary Fig. 2f). We thus speculate that the high levels of plasma inosine may be beneficial to ICB immunotherapy. However, the conclusion that inosine supplementation improves the response to ICB was not established until that inosine administration with ICB treatment significantly inhibited tumor growth in multiple mouse tumor models in our studies (Fig. 1c-e) and other lab's studies (Mager LF., et al. Science, 2020,

369:1481; Wang T., et al. Nature Metabolism, 2020, 2:635).

3) The authors use a mass spec study to support the binding of inosine to Uba6. Can they confirm direct binding in a cell free assay?

Response:

Thanks for your comment and suggestion. To directly address this question the reviewer raised, we set up and performed a new *in vitro* cell-free interaction assay that the purified UBA6 protein was incubated with inosine or vehicle before we added the purified USE1 (UBA6-specific ubiquitin E2) protein to identify whether inosine directly binds to UBA6 and potentially changes the structural conformation of UBA6 to impact its binding with USE1. Then, we used co-immunoprecipitation (Co-IP) and western blot to test the effect of inosine on interactions between the purified UBA6 and USE1 proteins. Indeed, the result showed that in comparison to the vehicle, the incubation of inosine with purified UBA6 protein remarkably reduced the subsequent interactions between the purified UBA6 and USE1 based on this cell-free assay *in vitro* (Supplementary Fig. 4e). As commented by the reviewer, we used chemical proteomics screening following a LiP-small molecule mapping (LiP-SMap) approach to detect proteins that become differentially susceptible to protease cleavage upon binding of inosine added to a proteome extract of 4T1 cells (Fig. 3a). Among 23 of 2470 proteins, UBA6 was one of the top hints that were identified to have structural changes induced by the binding of inosine (Fig. 3b, c), suggesting inosine directly binds to these proteins, especially UBA6. Additionally, we used the purified UBA6 and USE1 proteins to set up an *in vitro* biochemical assay for testing the direct effect of inosine on UBA6 activity. Our results showed that inosine inhibited the UBA6-mediated transfer of FAT10 in USE1 protein *in vitro* (Fig. 3f). Altogether, these results indicate that inosine inhibits UBA6 activity by directly binding to UBA6 protein.

4) Can the authors comment on the toxicity of inosine in their mouse models

Response:

We agree with the Reviewer that the toxicity should be addressed in our study. We have performed the toxicity experiments. No bodyweight loss was observed after treatment with Ino or Ino+ICB in the 4T1 model (Supplementary Fig. 3g). We also measured the serum levels of alanine aminotransferase (ALT) and aspartate aminotransferase (AST) to reveal no signs of hepatotoxicity

in the 4T1 tumor-bearing mice with Ino or Ino+ICB treatment (Supplementary Fig. 3h). Moreover, treatment with Ino or Ino+ICB did not induce a significant inflammation of lung and liver in 4T1 tumor-bearing mice (Supplementary Fig. 3i), confirming that inosine does not induce systemic inflammation. Based on all these measures utilized, we did not observe any adverse events associated with inosine therapy. Notably, inosine is widely used as a supplement and medicine, which people take inosine for improving their athletic performance and it is also used for multiple sclerosis and Parkinson's disease in clinical trials, confirming its safety for people in the clinic.

5) The introduction is relatively short and could be expanded.

Response:

We highly appreciate the reviewer's suggestions and comments. Following the suggestion of the Reviewer, we expanded the introduction to help give the reader a more detailed background regarding the metabolism, metabolites, and immunotherapy responses, and a full picture of why we initiated this project and how we ought to solve the knowledge gap of the interaction between metabolic dysfunction and immunotherapy responses (Line 56-77).

6) There are multiple grammatical errors that make the paper difficult to follow in places.

Response:

Thank the reviewer for raising this point for our attention which will help to improve the quality and readability of this paper. According to the reviewer's comments, we have invited a native English speaker to edit the manuscript.

Reviewer #3 (Remarks to the Author):

The report "Inhibiting tumor cell-intrinsic UBA6 by inosine augments tumor immunogenicity" Zhang et al. touches on a relevant topic in anti-tumor immunity. They show that inosine sensitizes different tumor cells to T cell mediated killing in vitro and in vivo, particularly when combined with immune checkpoint blockade therapy. This effect could be inhibited using broad spectrum antibiotics (orally), indicating that the effect may depend upon the microbiome. They further reveal that inosine induces this phenotype through UBA6. Complementing the in vitro and in vivo murine

studies they show that inosine is increased in the plasma of cancer patients responding to immune checkpoint blockade therapy. Lastly they also show that low UBA6 expression correlates with improved cancer patients survival and predicts the outcome of immune checkpoint blockade therapy.

Overall the study by Zhang et al. seems well performed and is relevant, particularly due to their dual approach (murine functional/mechanistic studies and validation in human samples/cohorts). However several key points need to be addressed before publication.

We appreciate the reviewer's nice comments recognizing our study “seems well performed and is relevant, particularly due to their dual approach”.

Major concerns:

1) The cancer cell direct effect of inosine is very intriguing, however the precise circumstances are unclear considering previous reports in the field. Wang, et al. Nat Metab. 2020 and Mager et al. Science 2020 have shown a direct effect of inosine on T cells for anti-tumor immunity. However, this report did not see an effect when immune cells were treated with inosine (Fig. 2a). How do the authors explain that? Could it be due insufficient activation of the T cells (only antigen stimulation without concurrent co-stimulation i.e. could anti-CD28 treatment change their findings in Fig. 2a). Moreover, Mager et al. Science 2020 describes inosine-tumor cell sensitization experiments similar to the experiments performed in this report and did not see an enhanced tumor cell death mediated by T cells. Is the direct effect of inosine on cancer cells dependent on the cell line? Does it only work on cell lines with no/low MHCI expression that is enhanced after inosine treatment, but not on cell lines with high baseline MHC expression?

Response:

We thank the reviewer for highlighting the relevant published reports in the field (Mager LF., et al. Science, 2020, 369:1481; Wang T., et al. Nature Metabolism, 2020, 2:635) showing a direct effect of inosine on T cells for anti-tumor immunity, which are quite interesting studies related to our research. Mager et al. (Science, 2020, 369:1481) use the MC38 model to do the *in vitro* T cell-mediated killing assay and the *in vivo* treatment experiments. Following up on the reviewer's suggestion, we have performed the T cell-mediated tumor cell killing experiment to evaluate the impact of inosine on MC38 cells. Inosine did not impact the proliferation and apoptosis of MC38 tumor cells and did not enhance inosine-pretreated MC38 tumor cell death mediated by T cells

(Fig. R1a, b), consistent with the literature (Mager LF., et al. Science, 2020, 369:1481). Based on our identified scenario of the inosine effect on tumor cells, UBA6 plays a critical role in the effect of inosine on T cell-mediated tumor killing (Fig. 3g) and ICB responses (Fig. 4, 5). We measured the protein levels of UBA6 on 4T1, B16-F0, B16-GMCSF, and MC38 tumor cells. Notably, the protein levels of UBA6 in 4T1 and B16-GMCSF tumor cells were higher than that in MC38 tumor cells (Fig. R1c). This result suggests that tumoral UBA6 expression level may determine the effect of inosine on these tumor cells' susceptibility to T cell killing. Given the similar baseline level of MHC-I expression on MC38 and 4T1 cells (Fig. R1d), we measured the stimulation gene pathway and found that inosine had a significant impact on the expression of antigen presentation related genes and inflammatory-related genes in 4T1 cells, but not MC38 cells (Fig. 2k, and Fig. R1e). The new evidence might explain the discrepancy of inosine impact on different tumor cells, confirming the hypothesis provided by the reviewer that the direct effect of inosine on cancer cells is likely dependent on the tumor cell line.

We respect and recognize the important findings in recent reports showing a direct effect of inosine on T cells for anti-tumor immunity (Mager LF., et al. Science, 2020, 369:1481; Wang T., et al. Nature Metabolism, 2020, 2:635). Compared to the ICB-resistant 4T1 model, the MC38 model has a more immunogenic microenvironment and is sensitive to anti-PD1 treatment. This feature is partially explained by their findings that the modest boosting effect of T cells by inosine has a synergetic effect in combination with ICB under CpG treatment, which boosts innate immunity to further activate antitumor T cells immunity (Mager LF., et al. Science, 2020, 369:1481). However, 4T1 and B16-GM models used in our study are more non-immunogenic and do not respond to ICB, even with combinational treatment of anti-CTLA4 and anti-PD1 to fully boost T cell immune response. But we found that inosine significantly enhanced ICB responses in the B16-GMCSF and 4T1 models without CpG treatment. Altogether, these findings indicate that the mechanism of action of inosine in the B16-GMCSF and 4T1 model might be different from that in the MC38 model *in vitro* and *in vivo*. Their studies and our work independently figure out the pleiotropic effects of inosine on antitumor immunity.

We recognized and cited these relevant reports and comments in the Discussion section (Line 359-366) by stating that “Aside from the effect of inosine on T cells, we surprisingly identified that the increased tumor cell immunogenicity also contributed to the function of inosine for driving antitumor immunity and enhancing current immunotherapy. The complementary mechanisms of inosine on tumor cells in combination with ICB targeting T cells reasonably explain the superiority of combinational therapy in multiple mouse tumor models. Thus, these findings in certain contexts indicate the pleiotropic effects of inosine on antitumor immunity by the complex and multiple action modes of the interactions between inosine and distinct components within tumor microenvironments.”

Figure R1: The impact of inosine on T cell-mediated tumor killing of MC38 cells.

a, The relative cell viability (left) and apoptosis (right) of MC38 cells following inosine treatment at indicated concentrations for 48h *in vitro* (n=5).

b, The relative cell viability of MC38-OVA cells was shown. OT-1 T cells were pretreated with indicated concentrations of inosine or vehicle for 24h, then co-cultured with MC38-OVA tumor cells at a 2:1 E: T ratio for 48h (n=5).

c, The protein level of Uba6 in various cancer cell lines was determined by western blot and the quantification of UAB6 protein was showed.

d, The mean fluorescence intensity of MHC-I on MC38 and 4T1 cell by flow cytometry.

e, The represented antigen processing/presentation and interferon-responsive gene expression in MC38 tumor cells treated with inosine at indicated concentrations (n=3).

2) The authors nicely demonstrate that inosine mediates its function through altered UBA6-FAT10- USE1 interactions. However, it is unclear how inosine does so. Does inosine change UBA6- FAT10- USE1 interactions through inducing upstream signalling and its consequences (i.e. through A1, A2A, A2B and A3 receptors) or through equilibrative nucleoside transporters (i.e. ENT1 and ENT2) and consequent changes in the cytoplasm. Could the authors block the effect of inosine with adenosine receptor inhibitors or inosine transport inhibitors (or genetic tools)?

Response:

The Reviewer raises the interesting question of how inosine implements its function on tumor cells. Although inosine has been reported to execute some of its functions by adenosine receptors (ARs) (He B., et al. J Exp Med, 2017; Welihinda AA., et al. Cell Signal, 2016), the expression levels of ARs in 4T1 tumor cells were low (Supplementary Fig. 5h) and ARs were also not observed by Lip-SMap approach in 4T1 cell lysate (Fig. 3b, and Supplementary Table S4), suggesting the effect of inosine on 4T1 tumor cells might not be mediated through ARs. To further reveal the mechanism of how inosine inhibits UBA6, we used adenosine receptors antagonist (CGS15943, 0.5 μ M) or inosine transport inhibitors (ENT1/ENT2 inhibitor: Dilazep dihydrochloride, 2.0 μ M) to pretreat 4T1 cells and measured the expression of a direct downstream gene signature of UBA6. As shown in Supplementary Fig. 5i, adenosine receptors antagonist had no impact on the inosine-induced expression of several UBA6-downstream genes. However, the inosine transport inhibitor significantly reversed the expression of these genes induced by inosine treatment. Collectively, these findings indicate that tumor cell-intrinsic proteins, especially UBA6, play a critical role in the effect of inosine on tumor immunogenicity.

3) The gut microbiome and specific bacteria thereof have been shown to improve immune checkpoint blockade therapy (also through the production of inosine). The authors own data using broad spectrum antibiotics (sFig. 1d-f) are in line with these observations. It would be important to at least profile the microbiome of the mice used in this study and evaluate whether certain bacterial taxa are enriched in mice treated with inosine and checkpoint blockade therapy compared to the other groups.

Response:

We highly appreciate the reviewer's suggestions and comments. We used 16S rRNA gene

sequencing to determine the microbiota composition in samples from the B16-F0 mouse model with Ctrl or ICB treatment. Consistent with prior reports in mice and humans (Routy B., et al. Science, 2018, 359:91; Vétizou M., et al. Science, 2015, 350:1079), the treatment with ICB induced gut microbiota dysbiosis in the B16-F0 mouse model (Supplementary Fig. 2a-d). Partial least squares discriminant analysis (PLS-DA) showed that the overall microbial community in B16-F0 tumor-bearing SPF mice with Ctrl treatment was completely separated from that in B16-F0 tumor-bearing SPF mice with ICB treatment (Supplementary Fig. 2a). Although a minor change in microbiota composition at the phylum and genus level in B16-F0 tumor-bearing SPF mice with ICB treatment, the relative abundance of some genera, such as *Parabacteroides*, *Akkermansia*, *Bifidobacterium*, were moderately changed by ICB treatment (Supplementary Fig. 2b-d). Notably, Mager et al. (Science, 2020, 369:1481) recently revealed that some species of *Akkermansia* and *Bifidobacterium* enhance the response to CTLA-4 antibody by producing large amounts of inosine. Collectively, these findings indicate that the inosine level alerted by ICB treatment may be at the least partially due to the microbiota dysbiosis induced by ICB treatment. However, we should recognize that the identity of indigenous bacteria related to ICB responses and overproducing inosine will be further explored.

4) The authors rely on published datasets to some extent. (i.e Fig. 1 and b, Fig. 3d, Fig. 6a and b, sFig.1 g and h, sFig. 3 b and c, sFig. a-f). While the benefit of these datasets is obvious, there are certainly some drawbacks. First example: Using plasma samples from cancer patients to detect inosine: While inosine is rather stable (half-life of 15 hours), adenosine- the precursor of inosine- is not (half-life of 10 seconds). Thus without tight control of the sampling procedure/duration and more importantly adenosine stabilization, the levels of inosine could drastically be altered. Second example: the proteomics approach to identify inosine modulated proteins (Fig.3a) identified 23 candidates. This was done in 4T1 breast cancer cells. UBA6 was then identified from these 23 candidates using a published database (Pan et al. Science. 2018), which relies however on B16F10 cells, a melanoma cell line. It is doubtful whether results in B16F10 can or should be used to draw conclusion from results obtained from 4T1 cells. While it is clear that the authors cannot repeat large human cohorts for their inosine measurements or transcriptomics analysis, the risks/limitations should certainly be discussed. However, the screen to identify UBA6 should be repeated by the authors with the same model system (4T1) in mini-screen targeting their 23

candidates.

Response:

We fully agree with the reviewer's comments. According to the reviewer's suggestions, we developed a pooled genetic screening approach to identify 23 genes that may increase or decrease the fitness of 4T1 tumor cells growing *in vivo*. After 12 days, we collected the tumors and compared the library representation in tumors from WT mice to tumors growing in NSG mice. Our results revealed that UBA6 deletion, among the 23 identified genes, had the highest negative score which indicates the increased sensitivity of tumor cells to immune attack *in vivo* (Supplementary Fig. 4d), consistent with the results (Fig. 3d and Supplementary Fig. 4b, c) from previous studies (Pan et al. Science, 2018, 359:770; Kearney et al. Science Immunology, 2018, 3:ear3451). Altogether, these findings demonstrate that tumor cell-intrinsic UBA6 plays a critical role in immunotherapy responses.

Regarding the limitations of our study that the reviewer mentioned, we agree with the reviewer that we are unable to validate and perform the inosine measurement or transcriptomics analysis in our cohort. Future studies attempting to further parse them are warranted. We pointed out the risks/limitations in the Discussion section (Line 384-388).

Minor concerns:

1) Gene nomenclature is not always consistent. This should be fixed. i.e. sFig. 6 legend (Uba6 vs UBA6)

Response:

We have now corrected this inconsistent regarding the gene nomenclature in the whole text, figures, and figure legends.

2) the authors show that UBA6 is downregulated across all cancer types and that low UBA6 expression is favorable also in BRCA (absence of immune checkpoint blockade therapy) (sFig6). However, other reports have associated low UBA6 with cancer progression. i.e. Liu et al. Oncotarget. 2017 showed that UBA6 suppresses EMT and cancer invasion. The role of UBA6 in cancer is still unclear and should be discussed more cautiously.

Response:

Thanks for pointing this out. We recognize that the detailed role of UBA6 in cancer is complicated and still undermined. We cited these references and addressed this important point in the Discussion section by stating that “UBA6 plays an important role in embryogenesis and multiple pathogenesises of diseases including cancer progression and metastasis, however, the impact of UBA6 on tumor-intrinsic immunogenicity has been never addressed before.” (Line 371-374).

3) the main results of the paper could be presented in more detail in the text to guide and help the reader a bit more through the key findings.

Response:

We thank the reviewer for bringing up this important point and apologize for not being precise enough on the presentation of the main results. We have presented the main results in more detail in the revised manuscript.

REVIEWERS' COMMENTS

Reviewer #2 (Remarks to the Author):

All of my original comments have been addressed

Reviewer #3 (Remarks to the Author):

The authors have addressed the raised criticism. Barring minor changes, it is this reviewer's opinion that this work is now suitable for publication. Mainly, some of the conclusions are rather vague and not meaningful. This could easily be addressed through minor changes in the text. Some examples are given below.

1) Figure R1 should be included in the main paper as a supplementary Figure. The authors should clearly discuss that their mechanism is cell line dependent and this cell line dependent effect is potentially due to baseline UBA6 expression. On that note it would be very nice to look at MC38 cell which overexpress UBA6 (i.e. lentiviral transduction) to solidify their conclusion in their response letter "This result suggests that tumoral UBA6 expression level may determine the effect of inosine on these tumor cells' susceptibility to T cell killing."

2) The reviewers have satisfactorily addressed the role of Adenosine receptors and equilibrative nucleoside transporters. The point of these experiments should be highlighted a bit more. Instead of "Together, our findings indicate that tumor cell-intrinsic proteins, especially UBA6, play a critical role in the effect of inosine on tumor immunogenicity." One could say that the effect of Inosine on UBA6 mediated tumor cell sensitization was dependent on ENT receptors on no on adenosine receptors.

3) I would not use the wording "gut microbiota dysbiosis". Dysbiosis is a very vague term and one could argue that tumor bearing mice are already dysbiotic even without ICB therapy. The authors could rather say "ICB therapy led to altered microbial composition in the gut following ICB therapy"

4) No further comments the mini screen in 4T1 tumor bearing mice is very nice and corroborated their previous data. The discussion part 384-388 is rather vague unfortunately. Please state specific limitations of the human cohort samples (i.e. impact of low half life of adenosine on your inosine measurements etc)

Responses to Reviewers' Comments

RE: NCOMMS-21-22862A

We would like to thank the editorial team and reviewers for the highly positive review of our revised manuscript. We are very glad that our revision satisfied Reviewer 2. Following the reviewers' suggestions, we have performed additional experiments and revised the manuscript accordingly to address all concerns of the reviewers. Here, we have listed our point-by-point replies to the additional comments raised by Reviewers 3. The reviewer's original comments were reproduced below in black for your easy reference.

Reviewer #2 (Remarks to the Author):

All of my original comments have been addressed.

Response: We thank the Reviewer for the positive assessment of our work.

Reviewer #3 (Remarks to the Author):

The authors have addressed the raised criticism. Barring minor changes, it is this reviewer's opinion that this work is now suitable for publication. Mainly, some of the conclusions are rather vague and not meaningful. This could easily be addressed through minor changes in the text. Some examples are given below.

Response: We appreciate the reviewer for providing constructive and insightful comments to guide us on further strengthening our manuscript.

1) Figure R1 should be included in the main paper as a supplementary Figure. The authors should clearly discuss that their mechanism is cell line dependent and this cell line dependent effect is potentially due to baseline UBA6 expression. On that note it would be very nice to look at MC38 cell which overexpress UBA6 (i.e. lentiviral transduction) to solidify their conclusion in their response letter "This result suggests that tumoral UBA6 expression level may determine the effect of inosine on these tumor cells' susceptibility to T cell killing."

Response: We highly appreciate the reviewer's excellent suggestions. Following the suggestions

of the reviewer, we have included the original Figure R1 in the main paper as a new supplementary Figure 6g-i. Then we discuss that these cell line-dependent effects of inosine on tumor immunogenicity are potentially due to baseline UBA6 expression. Moreover, following up on the reviewer's suggestion, we have performed new experiments to generate the *Uba6*-overexpressed MC38 cell lines and evaluated the impact of inosine on UBA6-overexpressed MC38 cells to T cell-mediated tumor cell killing. The new results clearly showed that inosine significantly increased the sensitivity of UBA6-overexpressed MC38 cells to T-cell mediated tumor cell killing, compared to control MC38 cells (new Supplementary Fig. 6j). Together, these findings suggest that tumoral UBA6 expression levels determine the tumor immunogenicity and contribute to the direct effect of inosine on tumor cells (Line 318-328).

2) The reviewers have satisfactorily addressed the role of Adenosine receptors and equilibrative nucleoside transporters. The point of these experiments should be highlighted a bit more. Instead of “Together, our findings indicate that tumor cell-intrinsic proteins, especially UBA6, play a critical role in the effect of inosine on tumor immunogenicity.” One could say that the effect of Inosine on UBA6 mediated tumor cell sensitization was dependent on ENT receptors on no on adenosine receptors.

Response: We fully agree with the reviewer’s comments. According to the reviewer’s suggestions, we clarified these experiments in detail in the revised manuscript. To highlight the important point of these experiments, we revised the statement in the text accordingly to explain the clearer mechanisms. These results suggest that the effect of inosine on tumor immunogenicity is likely dependent on intracellular inosine transported by ENT transporters, but not extracellular receptor ARs (Line 285-293).

3) I would not use the wording “gut microbiota dysbiosis”. Dysbiosis is a very vague term and one could argue that tumor bearing mice are already dysbiotic even without ICB therapy. The authors could rather say “ICB therapy led to altered microbial composition in the gut following ICB therapy”

Response: We agree with the reviewer’s comments. Following the reviewer’s suggestions, we revised the description in the text by stating “ICB therapy led to altered microbial composition in the gut in the B16-F0 mouse model following ICB therapy” in this revised manuscript (Line 108-110).

4) No further comments the mini screen in 4T1 tumor bearing mice is very nice and corroborated their previous data. The discussion part 384-388 is rather vague unfortunately. Please state specific limitations of the human cohort samples (i.e. impact of low half life of adenosine on your inosine measurements etc)

Response: Thanks to the reviewer for acknowledging the improvement of our newly added mini-screen experiments in 4T1 tumor-bearing mice. In terms of the specific limitations of human cohort samples, we fully agree with the reviewer's comments and revised the statement in the discussion section accordingly by discussing the impact of the low half-life of adenosine on inosine measurements (Line 413-416).